# An active mechanical Willis meta-layer with asymmetric polarizabilities

Yangyang Chen [1,4], Xiaopeng Li [1,4], Gengkai Hu[2], Michael R. Haberman [3] & Guoliang Huang [1✉]

Willis materials exhibit macroscopic cross-coupling between particle velocity and stress as well as momentum and strain. However, Willis coupling coefficients designed so far are intrinsically coupled, which inhibits their full implementation in structural dynamic applications. This work presents a means to eliminate these limitations by introducing an active scatterer in a mechanical meta-layer that exploits piezoelectric sensor–actuator pairs controlled by digital circuits. We experimentally demonstrate abilities of the Willis meta-layer, in beams and plates, for independently engineering transmission and reflection coefficients of flexural waves in both amplitude and phase and nonreciprocal wave propagations. The meta-layer is described by a flexural wave polarizability tensor, which captures independent higher-order symmetric-to-symmetric and symmetric-to-antisymmetric couplings. The active meta-layer is adaptive in real time for reconfigurable broadband operation thanks to its programmability. This work sheds a new light on unsurpassed control of elastic waves, ranging from vibration protections to ultrasonic sensing and evaluation of engineering structures.

[1] Department of Mechanical and Aerospace Engineering, University of Missouri, Columbia, MO 65211, USA. [2] School of Aerospace Engineering, Beijing Institute of Technology, Beijing 100081, China. [3] Walker Department of Mechanical Engineering, University of Texas at Austin, Austin, TX 78712, USA. [4]These authors contributed equally: Yangyang Chen, Xiaopeng Li. ✉email: huangg@missouri.edu

The primary goal of metamaterial design is to create wave-bearing materials that display properties and performance exceeding those of naturally occurring materials as a result of engineered microstructures. Metamaterial research has been active for over two decades and significant contributions span wide swaths of science and engineering, including electromagnetism[1–3], thermodynamics[4,5], acoustics[6–8], and elastodynamics[9–11]. Mechanical Willis materials exhibit coupling between particle velocity and stress as well as momentum and strain. In acoustics, this coupling has been an emergent effective material property that results from subwavelength asymmetry and/or long-range order[12], which is in agreement with predictions from homogenization theory in elastodynamics[13,14]. This intrinsic coupling, analogous to the bianisotropy in electromagnetism, has recently attracted significant attention in the context of acoustic and elastic metamaterials[15–27]. The stress–velocity and momentum–strain coupling offered by Willis materials provides appealing solutions in many applications including but not limited to perfect elastic wave cloaking[25], independent control of transmissions and reflections[17], perfect wavefront manipulations[19], and reciprocity breaking[12,16].

However, Willis coupling is usually very weak and negligible in most of passive mechanical and acoustic metamaterials. To overcome this, scatterers with strong local monopole-dipole coupling are introduced to metamaterials, which leads to observations of Willis coupling in acoustics[12,17–23,26], elastodynamics[13,14] and structural elements like elastic beams[24,27]. Unlike conventional metamaterials, passive linear Willis metamaterials are usually designed based on geometrically asymmetric locally resonant microstructures and therefore only yield non-negligible Willis coupling coefficients at narrow frequency bands. In addition, the coupling constants are intrinsically connected, posing fundamental constraints on applications for Willis materials such as broadband operation, violation of reciprocal propagation, and independent and non-conservative control of wave transmission and reflection[12,17].

The design and fabrication of active metamaterials/metasurfaces is an increasingly important area of metamaterial research. Active metamaterials/metasurfaces have been shown capable of demonstrating numerous interesting and superior properties and functionality, ranging from real-time tunability[28–32], broadband operability[33,34], and multifunctionality[28,35] beyond passive counterparts. Active acoustic Willis metasurfaces have opened doors to eliminate many of the physical constraints of the Willis coupling constants that are imposed by assumptions of passivity[12,23]. The connections between active metamaterials to homogenized Willis material parameters have recently been suggested in a few examples in the context of acoustics and spatiotemporally modulated materials with the main focus on nonreciprocity[12,16,23]. However, mechanical Willis media offer significant additional functionality due to the increased dimensionality of the parameter space associated with its ability to support significantly more complex wave motion. To date, research on the topic of mechanical/elastic Willis media has only touched the surface of what is possible, and many properties and functions are still to be discovered, especially when active elements are considered. Specifically, the ability to design active mechanical Willis scatterers that provide arbitrary control of the scattering polarizabilities is one of the fundamental challenges in further improving the ability to manipulate wavefronts over wide frequency ranges.

In this work, we present exploration of the ability of an active mechanical Willis meta-layer to realize independent control of the transmission and reflection of flexural waves in beams and plates and break reciprocity (Fig. 1). This is achieved using independently activated symmetric and antisymmetric scattered fields, such that the wave fields in the left and right sides of the

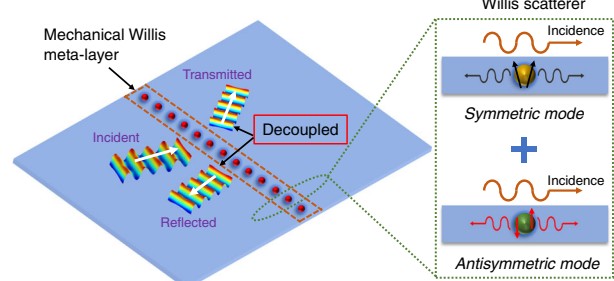

**Fig. 1 Schematic of the mechanical Willis meta-layer embedded with an array of active Willis scatterers.** The reflected and transmitted fields can be independently controlled by the active Willis scatterers. The active Willis scatter with asymmetric polarizabilities can operate symmetric and antisymmetric modes.

meta-layer can be arbitrarily manipulated (Fig. 1). An active Willis meta-layer is then designed and fabricated by introducing piezoelectric sensor–actuator pairs controlled with digital circuits, where electrical transfer functions are encoded. The active meta-layer is shown to efficiently and precisely control targeted modes by building scatterers that demonstrate unidirectional symmetric–symmetric and symmetric–antisymmetric coupling. To demonstrate the wide breadth of potential future applications, we use this approach to create a perfect wave absorber, non-reciprocal wave transmission, and independent wavefront manipulation in transmitted and reflected domains. Experimental observations of a perfect absorbing layer and nonreciprocal wave response due to the introduction of an active meta-layer in a beam are interpreted through theoretical analysis of the system as an effective Willis scatterer represented by an effective polarizability tensor lacking major symmetries. Finally, source-driven homogenization theory is employed to describe a mechanical Willis medium with independently controlled coupling coefficients for a beam consisting of an array of active Willis scatters. The Willis meta-layer presented here exploits the local mechanical response and active elements to enable compact and reconfigurable broadband wave manipulation which greatly surpasses what is achievable using passive materials, making it an excellent candidate for low-frequency vibration and wave control.

## Results

**Microstructural design and working principles.** To construct the mechanical Willis meta-layer, an array of thin slits is first cut out of a host plate such that one sensing and two actuating beams are formed in each of the meta-atoms ("scatterers") as shown in Fig. 2a. One piezoelectric sensor is bonded at the center of the sensing beam and two pairs of piezoelectric actuators are attached on the actuating beams. The actuators are placed symmetrically with respect to the sensor (Fig. 2a). The sensor is used to detect the incident wave by measuring the local curvature on the sensing beam, and actuators are employed to generate desired symmetric and antisymmetric scattered fields. The sensor and actuators are connected to a digital control system. The two actuators in the left share the same output voltage $V_{a1}$, and the two actuators in the right share the same output voltage $V_{a2}$ (Fig. 2a). It should be mentioned that, once actuating, the sensing signal contains information of incident waves and feedback responses produced by the actuators. To eliminate those feedback sensing components due to actuating, a transfer function, $G$, defined as feedback responses of the mechanical system is implemented to the actuating units (See Supplementary Note 1 for detailed

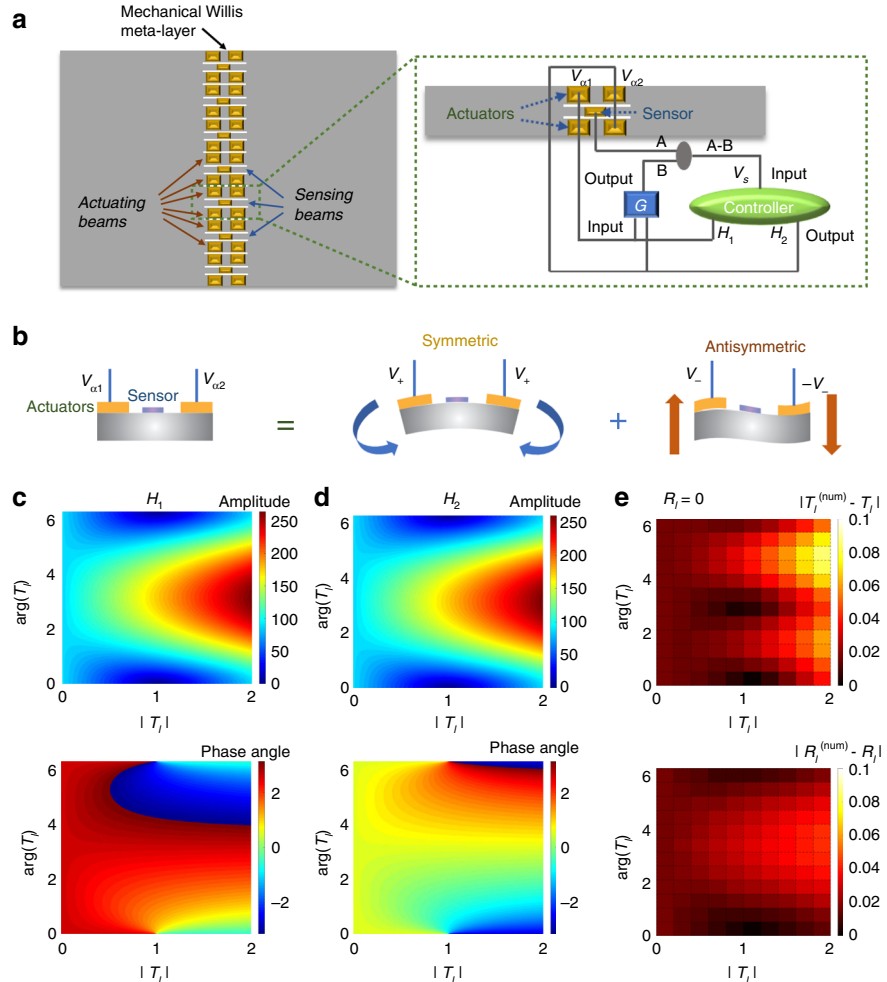

**Fig. 2 Design and working principles of the Willis meta-layer. a** Microstructural design of the active Willis meta-layer. Geometric and material parameters of the design are given in Supplementary Fig. 4 and Supplementary Table 2. **b** Working principles of the active meta-layer. **c, d** Designed amplitudes and phases of electrical transfer functions, (**c**) $H_1$ and (**d**) $H_2$ of a transmission-type meta-layer ($R_l = 0$) with the amplitude of the transmission coefficient, $T_l$ varied from 0 to 2 (horizontal axis) and with a full $2\pi$ phase change coverage (vertical axis). Source data are provided as a Source Data file. **e** Differences between numerically calculated transmission and reflection coefficients and their prescribed values to be achieved with the corresponding transfer functions in (**c**) and (**d**). Source data are provided as a Source Data file.

descriptions of $G$). The output from $G$ is subtract from the sensing voltage originating from the sensor to constitute a new sensing signal, $V_s$, representing the incident signal and as the input to the controller (Fig. 2a), creating a feedforward control loop. This is the key to system stability. The circuit design is detailed in Supplementary Note 2.

Transfer functions, $H_1 = V_{a1}/V_s$ and $H_2 = V_{a2}/V_s$, are encoded in the controller and control relations between sensing and actuating signals of the active Willis meta-layer. Specifically, the symmetric part of actuating voltages $V_+ = (V_{a1} + V_{a2})/2$ produces an effective bending curvature that generates the symmetric wave mode as illustrated in Fig. 2b. On the other hand, the antisymmetric part of actuating voltages $V_- = (V_{a1} - V_{a2})/2$ induces an effective shear strain that produces the antisymmetric wave mode (Fig. 2b). Therefore, combining the symmetric and antisymmetric parts of transfer functions provides a simple and efficient way to independently tailor transmission and reflection coefficients of flexural waves with the incidence from one side of the Willis meta-layer. In other words, transfer functions $H_1$ and $H_2$ can be determined for any desired transmission and reflection coefficients for the incidence from the left side, $T_l$ and $R_l$, as (see

Supplementary Note 3 for details)

$$H_1 = \frac{(T_l - 1)e^{ika} - R_l e^{-ika}}{\kappa_s \kappa_a e^{2ika} - \kappa_s \kappa_a e^{-2ika}},$$
$$H_2 = \frac{(T_l - 1)e^{-ika} - R_l e^{ika}}{\kappa_s \kappa_a e^{-2ika} - \kappa_s \kappa_a e^{2ika}},$$
(1)

where $a$, $k$, $\kappa_s$, and $\kappa_a$ denote the horizontal distance between the sensor and the actuator, wavenumber of flexural waves in the beam, electromechanical coupling coefficients of the sensor and actuator, respectively. Note that passive scattering due to inhomogeneities in materials and geometries are ignored as they are of significantly lower amplitude than the actively scattered fields, due to the deep subwavelength geometry of sensors and actuators (see Supplementary Note 4 for details).

To examine the performance of the design on independent control of transmitted and reflected fields, numerical simulations of flexural wave interaction with a Willis meta-layer atom are conducted. Figure 2c–e shows the design of a transmission-type meta-layer ($R_l = 0$) with the amplitude of the transmission coefficient, $T_l$ varied from 0 to 2 (horizontal axis) and with a full

$2\pi$ phase change coverage (vertical axis). Figure 2c, d shows amplitudes and phases of electrical transfer functions, $H_1$ and $H_2$, required to achieve this control (see Supplementary Note 3 for details). It is found that $H_1$ and $H_2$ possess the same amplitudes and $\pi/2$ phase differences. To quantitatively measure the success of the design, Fig. 2e shows differences between the numerically determined transmission, $T_l^{(num)}$ and reflection, $R_l^{(num)}$ coefficients and their desired values, $T_l$ and $R_l$, that the meta-layer is designed to achieve ($|T_l^{(num)} - T_l|$ and $|R_l^{(num)} - R_l|$). It can be found that the Willis meta-layer can accurately reach the desired transmission and reflection coefficients with small absolute errors, and the largest errors are found at points of high transmission. Note that the meta-layer also enables manipulation of the reflected field, and changing the functions of the Willis meta-layer, i.e. from the transmission-type to reflection-type, only requires reprogramming electrical controllers.

Furthermore, we also find that the transmission and reflection coefficients for the incidence from the right side, $T_r$ and $R_r$, are different from those with the incidence from the left side with the transfer functions given in Eq. (1) and their relations can be expressed as (see Supplementary Note 3 for details)

$$T_r = 1 + R_l,$$
$$T_l = 1 + R_r. \qquad (2)$$

The Willis meta-layer is therefore intrinsically nonreciprocal.

**Willis meta-layer on a beam**. The unique properties of the Willis meta-layer can circumvent fundamental constraints of extreme flexural wave control with conventional approaches. For example, designing deep subwavelength structures for broadband one-way total absorption of flexural waves in the transmitted domain has been extremely challenging or even impossible, since high material damping properties and impedance match conditions are required at the same time[36,37]. However, with the current approach, total absorption can be implemented through properly programming electrical transfer functions to induce $T_l = R_l = 0$ (or $T_r = R_r = 0$ by taking a mirror operation). To demonstrate this phenomenon, experiments on a beam with the Willis meta-layer are conducted. Figure 3a shows the fabricated sample and experimental setup of the test (see Supplementary Note 5 for details). The Willis meta-layer is placed in the middle of a host steel beam and two piezo-electric actuators are attached to the left and right sides of the meta-layer to generate incident flexural waves from each direction. A ten-cycle tone-burst signal with a central frequency of 10 kHz is used for this measurement. The out-of-plane velocity wave field is measured by a scanning laser Doppler vibrometer (Polytec PSV-400). Figure 3b, c shows experimentally measured wave fields with incidences from both sides, when the control circuit is switched ON and OFF, respectively. Comparing left portions of these figures where the incidence is from the left side, nearly perfect absorption of flexural waves is clearly seen, when the circuit is switched ON to induce the desired function of the Willis meta-layer. Whereas the wave can pass through the meta-layer, when the circuit is switched OFF. Passive wave scattering due to the presence of the meta-layer is small enough to be ignored. To examine nonreciprocity, we change the incident source from the left to the right side of the meta-layer (Fig. 3b, c). As can be observed from these figures, the transmitted wave fields are almost same for the cases with the circuit switched ON and OFF, indicating that the Willis meta-layer displays unitary transmission for incidence from right. In addition to the unitary transmission, we also notice that the reflected wave shown in the bottom of Fig. 3c has the same amplitude as that of the incident wave, such that the Willis meta-layer functions as a perfect transparent mirror for incidence from the right. This extreme nonreciprocal behavior is easily found using Eq. (2), where

zero transmission and reflection for one side of incidence will induce unitary transmission and reflection for the other side incidence. Corresponding 3D piezoelectric-coupled numerical simulations are also performed to validate the phenomena, and good agreement is observed as shown in Fig. 3d.

Importantly, the programmable Willis meta-layer is not limited to narrowband operation, rather, it is a reconfigurable system to tailor the desired reflection, transmission, and absorption response for broadband applications. To achieve this, we first study the frequency responses of electromechanical coupling coefficients of the sensor and actuators by performing numerical and experimental tests and transfer functions are then formulated based on Eq. (1). Note that these transfer functions are derived from frequency-domain analyses, and therefore causality may not be guaranteed in time domain. The Supplementary Note 3 provides details of the causal transfer functions implemented in experiments reported here. As an illustration, we reconfigure the controller to achieve the one-way perfect absorber from 9 to 18 kHz. Figure 3e shows the transmittance, reflectance and absorption of the Willis meta-layer for incidence from left and right sides for both numerical simulations and experiments. It can be clearly seen that for incidence from the left side, both the transmittance and reflectance are almost zero and absorption is around one in the designed frequency band. The Willis meta-layer behaves as a perfect absorber for this case. Whereas, when incidence is from the right, both the transmittance and reflectance approaches unity and absorption approaches negative one in the designed frequency band. For this direction of incidence, the Willis meta-layer behaves as a perfect transparent mirror. Furthermore, it is important to mention that functionalities provided by the programmable Willis meta-layer can be tuned in real time by coding the controller, such that switching between perfect absorption, total transmission and perfect transparent mirror can be executed in seconds and even adjusted due to changing conditions. The reconfigurable broadband response and tunability are both significant improvements over what are possible using passive Wills elements.

**Polarizability tensors for flexural waves**. To understand the behavior observed in the experiments described above and facilitate interpretation of the meta-layer as a Willis scatterer, we develop an analytical model based on polarizability tensors. The polarizability tensors describe general scattering properties of the Willis meta-layer in the beam, which is considered as a point scatterer for flexural waves. Analogous to previous works in electromagnetism and acoustics[22], we formulate the polarizability tensor, **β**, for flexural waves as

$$\tilde{\mathbf{Q}} = \boldsymbol{\beta}\mathbf{F}_{\mathbf{loc}}, \qquad (3)$$

where $\mathbf{F}_{\mathbf{loc}} = [\psi_{loc} \quad w_{loc} \quad F_{loc} \quad M_{loc}]^T$ is the local wave field vector at the scatterer location, with $\psi_{loc}$, $w_{loc}$, $F_{loc}$, and $M_{loc}$ denoting local rotational angle, transverse displacement, shear force and bending moment, respectively. The vector $\tilde{\mathbf{Q}} = [\tilde{q}_0 \quad \tilde{f}_0 \quad \tilde{s}_0 \quad \tilde{p}_0]^T$ is the multipole vector, representing the "excited" (scattered) field caused by the interaction of the point scatterer with the local field, with $\tilde{q}_0$, $\tilde{f}_0$, $\tilde{s}_0$, and $\tilde{p}_0$ representing the body torque, transverse body force, shear strain and bending curvature, respectively. The Supplementary Note 6 provides a discussion and visualization of the scattered fields associated with the multipole vector. In particular, $\tilde{f}_0$ is a monopole quantity of order zero in the multipole expansion and $\tilde{q}_0$ and $\tilde{s}_0$ are dipole quantities of order one. Similarly, $\psi_{loc}$, $w_{loc}$ and $F_{loc}$ are dipole, monopole, and dipole quantities, respectively. Whereas, $\tilde{p}_0$ and $M_{loc}$ are longitudinal quadrupole quantities. In a beam, $w_{loc}$, $M_{loc}$, $\tilde{f}_0$ and $\tilde{\varphi}_0$ are related to symmetric modes, where outward

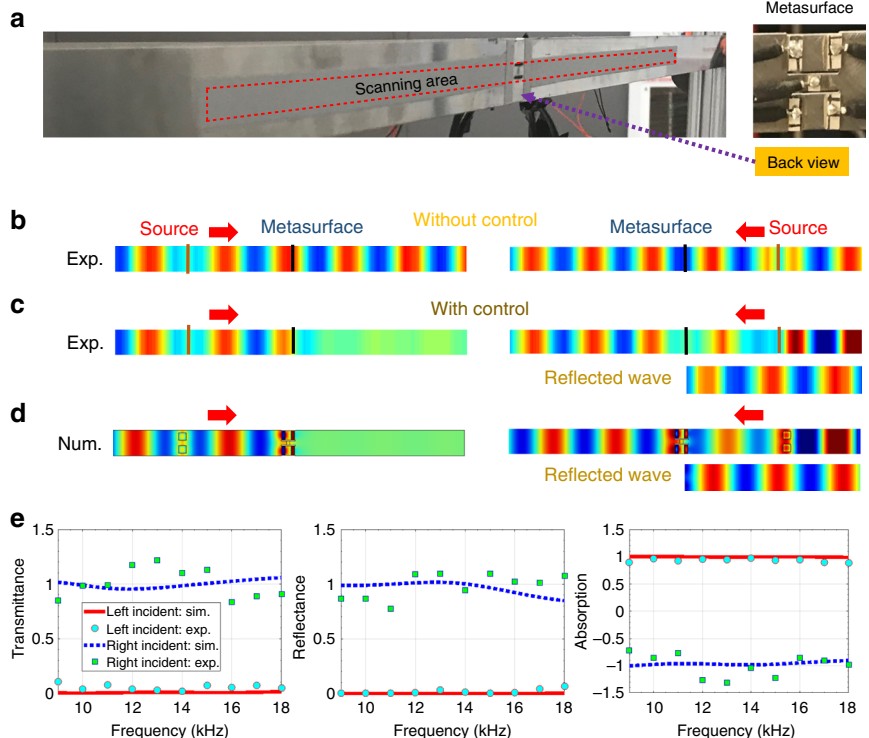

**Fig. 3 Experimental demonstrations of the Willis meta-layer in beams. a** Schematic of the experimental setup. Parameters of experimental setup are given in Supplementary Fig. 4 and Supplementary Table 2. **b** Experimentally measured wave field without control (circuit is switched OFF). Source data are provided as a Source Data file. **c** Experimentally measured wave field with control (circuit is switched ON). Source data are provided as a Source Data file. **d** Corresponding numerical simulations to validate the results in (**c**). **e** Experimentally measured and numerically simulated transmittance, reflectance and absorption for frequencies from 8 to 18 kHz demonstrating the reconfigurable broadband operability. Source data are provided as a Source Data file.

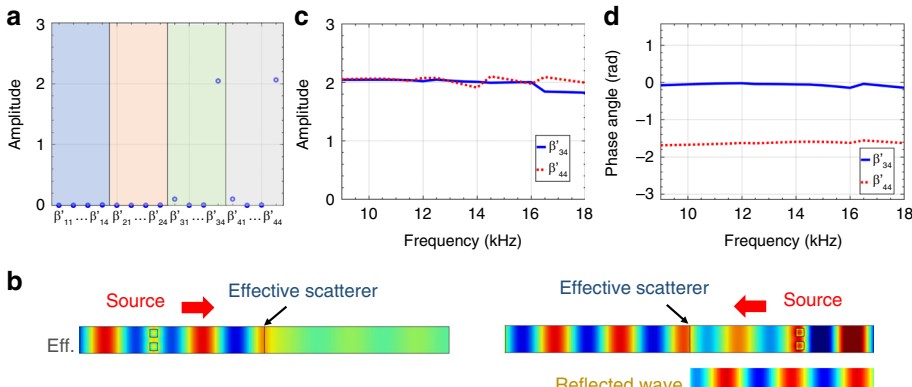

**Fig. 4 Effective polarizabilities of the Willis meta-layer. a** Numerically retrieved amplitudes of the 16 normalized polarizabilities using the wave fields calculated in Fig. 3a at 10 kHz. Source data are provided as a Source Data file. **b** Numerically simulated wave fields at 10 kHz using the effective model characterized by the polarizabilities in Fig. 4a. (**c, d**) Numerically retrieved (**c**) amplitudes and (**d**) phase angles of normalized nonzero polarizabilities using the wave fields calculated in Fig. 3e. Source data are provided as a Source Data file.

propagating waves are in-phase when traveling in opposing directions, and $\theta_{\mathrm{loc}}$, $F_{\mathrm{loc}}$, $\tilde{m}_0$ and $\tilde{\psi}_0$ are related to antisymmetric modes, where outward propagating waves are out-of-phase when traveling in opposing directions. It is also important to mention that the scattering properties of the Willis scatterer presented here cannot be properly captured using the simple Euler beam theory because it lacks the degrees of freedom associated shearing motion within the beam. We therefore employ Timoshenko beam theory.

To obtain all the 16 polarizability coefficients of the Willis meta-layer in the 4×4 matrix, $\boldsymbol{\beta}$, we have formulated the polarizability-retrieval method with Timoshenko beam assumptions by considering both propagated (far-field) and evanescent (near-field) wave solutions (see Supplementary Note 7 for details). Figure 4a shows the amplitudes of those retrieved normalized complex polarizabilities of the Willis meta-layer demonstrated in Fig. 3d. We find $|\beta'_{34}|$ and $|\beta'_{44}|$ are close to 2, and other polarizabilities are near zero, making the polarizability tensor asymmetric (see Supplementary

Note 7 for definitions of $\beta'_{34}$ and $\beta'_{44}$). In particular, $\beta'_{34}$ describes the coupling between the quadrupole bending moment and the dipole shear strain, and $\beta'_{44}$ represents the coupling between the quadrupole bending moment and the quadrupole bending curvature. Thus, $\beta'_{34}$ and $\beta'_{44}$ can be termed as quadrupole-to-dipole and quadrupole-to-quadrupole polarizabilities, respectively. The retrieved polarizabilities are also validated by numerical simulations where an effective scatterer characterized by those retrieved polarizabilities is implemented into a beam (Fig. 4b). It is clearly seen that the wave fields calculated are almost the same with those in Fig. 3c, d where the meta-layer is physically implemented, demonstrating validity of the polarizability model. Figure 4c, d show amplitudes and phase angles of the two nonzero normalized polarizabilities, $\beta'_{34}$ and $\beta'_{44}$, of the Willis meta-layer demonstrated in Fig. 3e for the frequencies from 9 to 18 kHz, respectively. It is found that amplitudes of $\beta'_{34}$ and $\beta'_{44}$ are around 2, and phase angles of $\beta'_{34}$ and $\beta'_{44}$ are near 0 and $-\pi/2$, respectively. Furthermore, it should be noted that Willis coupling reported in acoustics and elastodynamics generally only considers monopole-dipole coupling. However, the Willis coupling in the meta-layer presented in this work exploits coupling between higher order terms, specifically quadrupole-to-dipole and quadrupole-to-quadrupole couplings.

To find insights of the two nonzero polarizabilities, we rewrite transmission and reflection coefficients in terms of $\beta'_{34}$ and $\beta'_{44}$ as

$$T_l \approx 1 - \frac{\beta'_{34} + i\beta'_{44}}{4}, \quad R_l \approx \frac{\beta'_{34} - i\beta'_{44}}{4},$$
$$T_r \approx 1 - \frac{-\beta'_{34} + i\beta'_{44}}{4}, \quad R_r \approx -\frac{\beta'_{34} + i\beta'_{44}}{4}. \tag{4}$$

Inserting values of $\beta'_{34}$ and $\beta'_{44}$ from Fig. 4c, d into Eq. (4), the derived transmission and reflection coefficients coincide with the wave phenomenon in Fig. 3e. In addition, we notice that, in the presence of $\beta'_{34}$, reciprocity is broken since $T_l$ is no longer equal to $T_r$. This is because the asymmetry of the polarizability tensor for the active meta-layer only permits symmetric-to-antisymmetric scattering and not the converse, i.e. $\beta'_{34} \neq 0$ but $\beta'_{43} = 0$. We also note that the local bending moments are identical no matter the direction of incidence, which results in the same induced shear strain, $\tilde{s}_0 = \beta_{34} M_{loc}$. This shear strain excites antisymmetric modes propagating to left and right, such that the final wave interference patterns are different for incidences from the left or right sides. Furthermore, for a given direction of incidence, the corresponding transmission and reflection coefficients can be arbitrarily and precisely tailored using Eq. (4) by properly designing polarizabilities, $\beta'_{34}$ and $\beta'_{44}$. One can therefore combine the scattered symmetric and antisymmetric modes excited by the bending curvature and shear strain to produce any wave interference pattern in transmitted and reflected domains. The reflected and transmitted fields generated using this approach can be controlled independently and simultaneously keeping in mind that the transmission and reflection coefficients for the incidence from the other direction will be automatically determined, as indicated by Eqs. (2) and (4). In fact, electrical transfer functions and mechanical polarizabilities are intrinsically related by the simplified expressions $\beta_{44} = \frac{\chi_q(H_1 + H_2)}{2}$ and $\beta_{34} = \frac{\chi_f(H_1 - H_2)}{2}$, where $\chi_q$ and $\chi_f$ denote factors describing the electromechanical coupling of symmetric and antisymmetric modes, respectively.

**Willis meta-layer in plates**. We next investigate an active Willis meta-layer consisting of 10 unit cells on a steel plate as shown in Fig. 5a to demonstrate the ability to independently control flexural wave fields in the transmitted and reflected domains. The

design presented here considers the case where each unit cell is connected to their own unique circuit and is controlled independently. An array of piezoelectric actuators is bonded to the left of the meta-layer to generate incident plane flexural waves and we again use a ten-cycle tone-burst signal with the center frequency of 10 kHz for the experimental demonstration. The unit cells of the meta-layer are first programmed to achieve absorption for incidence from the left (Fig. 5b). The measured out-of-plane velocity fields for this case are shown in Fig. 5c, where near total absorption of flexural waves is clearly observed. Numerical simulations using the same geometric and material parameters show good agreement (Fig. 5d). Note that the performance of the meta-layer is not restricted to plane-wave incidence. For example, excellent absorption performance is observed for cylindrically spreading wave incidence due to point excitation (See Supplementary Fig. 9).

Furthermore, the programmable Willis meta-layer is also able to support wavefront redirection. By programming a linear phase profile for the transmission coefficient, $T_l$, along the meta-layer and enforcing $R_l = 0$, the transmitted wave can be steered in any desired direction while negating the reflected field (Fig. 5e–g). Similarly, by enforcing $T_l = 0$, the reflected wavefront can be controlled by specifying the phase profile applied to the meta-layer to generate a position-dependent reflection coefficient, $R_l$ (See Supplementary Fig. 10). Beyond the conventional applications such as perfect absorption or arbitrary control of reflected or transmitted wavefronts, the programmable Willis meta-layer can transform both the transmitted and reflected wavefronts at the same time by prescribing phase profiles on $T_l$ and $R_l$ simultaneously (Fig. 5h–j). The example shown in the figure is the result of encoding a hyperbolic phase profile in $T_l$ to focus the transmitted wave, whereas a linear profile is encoded in $R_l$ to steer the reflected wave to a specified direction (24° from normal). The out-of-plane wave field shown in this figure demonstrates the successful operation of the meta-layer to display independent control of transmissions and reflections.

**Homogenization of a beam with periodic active scatterers—a Willis beam**. In addition to the Willis meta-layer studied above, it is also of scientific importance to construct a new 1D Willis medium by using a periodic arrangement of Willis meta-layers (scatterers) on a beam[27] (see Supplementary Notes 9 and 10 for details). Effective properties of the Willis medium are analytically formulated as a function of local polarizabilities. It is interesting to note that the effective mass density, rotational inertia and shear compliance of the beam with the scatterers presented here are left unchanged, and only one Willis coupling coefficient is nonzero. Applying conservation of translational and rotational momentum, we find that the nonzero coupling coefficient will induce non-reciprocal wave propagation in the periodic Willis beam considered in this work. On the other hand, the effective bending stiffness of the Willis beam is modified by the polarizability, $\beta_{44}$, which is reciprocal. This description provides a theoretical foundation for the creation of nonreciprocal 1D and 2D Willis media to provide unprecedented control of elastic waves in structural elements like beams and plates by using active scatterers.

**Discussion**
In summary, we have provided a theoretical and experimental study of an active Willis meta-layer using a sensor–actuator control loop to realize independently controlled asymmetric polarizabilities. It has been shown that this approach results in the independent control of transmitted and reflected wave fields in beam and plate structures and that the system is nonreciprocal. Theoretical predictions have been experimentally demonstrated and are in

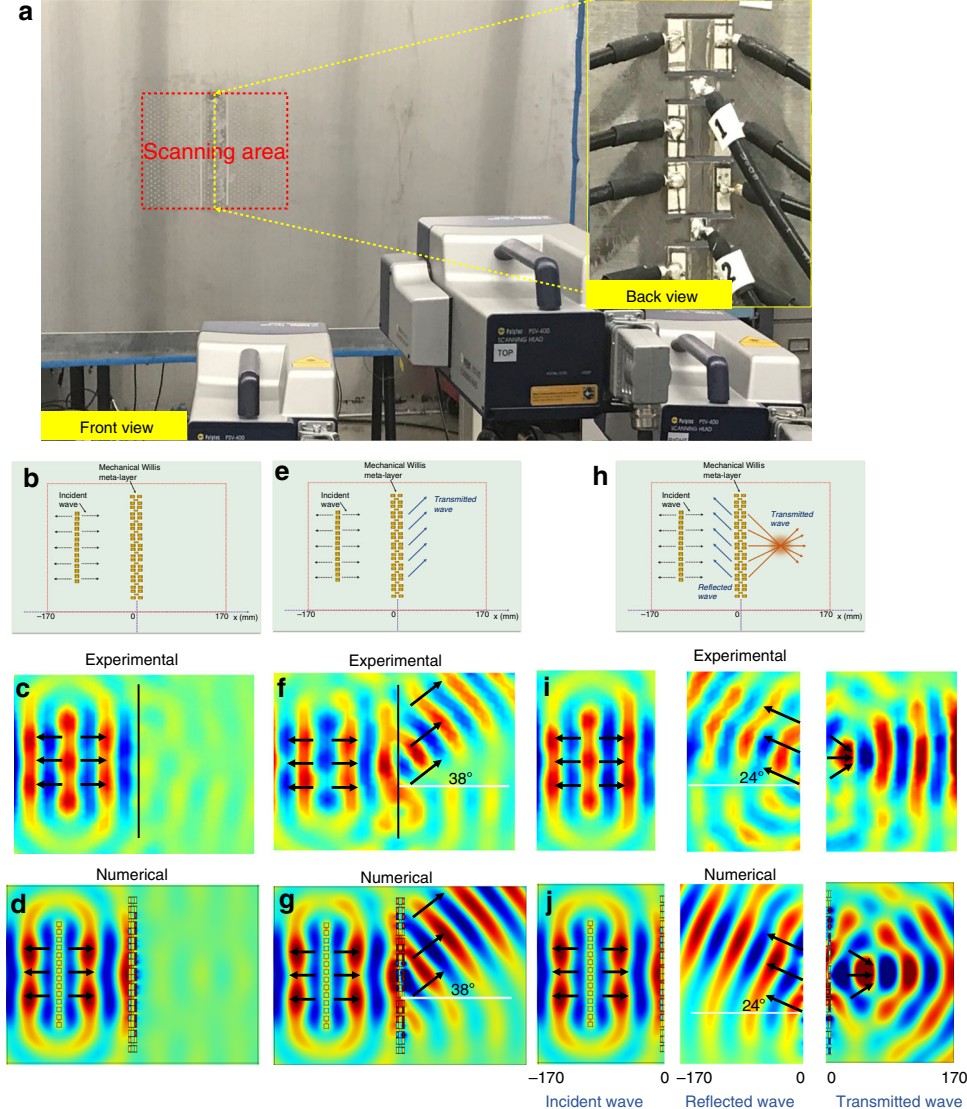

**Fig. 5 Experimental demonstrations of the Willis meta-layer in plates. a** Image of the experimental setup of the flexural wave propagation through a Willis meta-layer in a plate with the Willis meta-layer, and reflected and transmitted regions highlighted. **b–j** Experimental and numerical demonstrations of the Willis meta-layer for (**b-d**) total wave absorption, (**e-g**) wave steering in the transmitted domain and (**h-j**) simultaneous wave focusing and steering in both transmitted and reflected domains, respectively. Source data are provided as a Source Data file.

excellent agreement with numerical simulations. The proposed Willis meta-layer can be programmed in real time on demand and reconfigurable at broadband frequencies. This approach is a significant advancement over passive systems that are inevitably narrowband and cannot easily be tuned. Active Willis meta-layers in plates and beams therefore provide a new approach to design future unsurpassed elastic wave control devices.

## Methods

**Numerical simulations**. Three-dimensional numerical simulations of harmonic wave propagations through the Willis meta-layer are conducted using the commercial finite element software, COMSOL Multiphysics. In numerical simulations, the 3D linear piezoelectric constitutive law is applied to the piezoelectric patches. The sensing signal is obtained by integrating free charges over a surface of an electrode of the piezoelectric sensor. The two electrodes on the piezoelectric sensor have zero electric potential. The incident flexural wave is generated by applying a harmonic voltage across a piezoelectric patch bonded to the host beam on the left- and right-hand sides of the meta-layer. Two perfectly matched layers (PMLs) are attached to both ends of the host beam in order to suppress reflected waves from the boundaries. Two displacement probes are defined on the host beam in the right- and left-hand sides of the meta-layer to measure the out-of-plane displacement and calculate wave transmission and reflection coefficients.

## Data availability

All the data supporting the findings of this study are available from the corresponding authors upon reasonable request. Source data are provided with this paper.

## Code availability

The computer code and algorithm that support the findings of this study are available from the corresponding author upon reasonable request. Source data are provided with this paper.

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

## Acknowledgements

The authors gratefully thank Prof. Hussein Nassar from University of Missouri for valuable discussions. This work is supported by the Air Force Office of Scientific Research under Grant No. AF 9550-18-1-0342 with Program Manager Dr. Byung-Lip (Les) Lee and the NSF EFRI under award No. 1641078. M.H. acknowledge additional support from ONR YIP Grant No. N0014-18-1-2335. G.K.H. acknowledge support from the National Natural Science Foundation of China Grant No. 11632003.

## Author contributions

Y.C. and G.L.H. (Guoliang Huang) proposed the concept; X.L. conducted experiments; Y.C. performed theoretical and numerical investigations; Y.C., M.H., G.K.H. (Gengkai Hu), and G.L.H. discussed modeling and wrote the manuscript; G.L.H. supervised the research; All authors interpreted the results and reviewed the manuscript.

## Competing Interests

The authors declare no competing interests.
