## [Peer Review File · Nature Communications]

Reviewers' comments:

Reviewer #1 (Remarks to the Author):

This manuscript presents an active elastic wave metamaterial, based around a sensor which measures the symmetric component of the incident field, and is then able to generate controlled levels of both symmetric and anti-symmetric waves. This is presented as an example of non-reciprocal Willis coupling in a metamaterial, and this description is quite sound.

However, the authors model this system with what they call a "polarizability" model, attributing the symmetric radiation to excitation of a monopole mode, and anti-symmetric radiation to a dipole mode. This terminology does not seem well justified, for example the bending moment M is a higher order moment of the stress tensor, so describing it as a monopolar term is quite confusing. We should expect the degree of a multipole to correspond to some order of near-field spatial variation, or to a far-field radiation pattern.

Furthermore, the authors first introduce a 4x4 matrix description, with 2 "dipole" and 2 "monopole" quantities in both source and induced quantities, but quickly reduce to only considering a 2x2 matrix, without any real justification. It is quite possible that the other quantities do play a role in the experiment, but given the number of observable transmission/reflection coefficients, it is not possible to resolve all 16 coefficients. It would be much to simplify the model to describe symmetric and anti-symmetric components, without making claims about multipole orders, or unnecessarily trying to differentiate indistinguishable scattering mechanisms.

Overall this theoretical approach is quite difficult to follow, and seems unnecessary to describe the experiment. For example, the required voltage transfer functions given by Eq (24) in the supplementary material do not require the polarizability model, but just depend on simple quantities such as required transmission, separation between transducers etc. In particular, the source driven homogenization does not seem to relate to the experimental part, or to make any new predictions, and homogenization is not meaningful for the considered cases of a single scatterer and an inhomogeneous line array of scatterers.

On the other hand, the experimental aspects of this paper seem novel, well conducted, and quite interesting. In my view, this part of the paper clearly meets the level of novelty I would expect of Nature Communications, and could readily be understood by an abroad audience. If the experimental results were given more prominence, and the theoretical part was reduced to the minimum necessary to interpret the experiments, this would be quite a solid paper.

I note a few minor issues:

- In several places numerical simulations are performed, however it is never described how these are done. It appears that some form of full wave elastic and piezoelectric coupled model is used.
- On line 38 the statement "furnish unsurpassed control" seems like hype, and should be replaced with something more substantive
- On lines 49-50, vague mention is made of the constraints of passive media for certain applications. This claim should be made more concrete and backed up by examples.
- On lines 193-194, it is stated that the scattering in the passive state is negligible. This statement should be quantified
- On line 266, reference titles suggest that reference [21] should have been cited instead of [20]

Reviewer #2 (Remarks to the Author):

Review of NCOMMS-19-35700

The broad context of the work is Willis materials—a class of mechanical metamaterials with unusual dynamic constitutive relations, namely, velocity-dependent stress and strain-dependent linear momentum. The authors propose and experimentally realize a structural counterpart of Willis materials, by considering beams undergoing flexural motions with analogous cross-coupling. To my knowledge, the novelty in this work is the implementation of active elements in the settings of Willis beams. As a result of the integration of these elements, the authors demonstrate (theoretically, numerically, and experimentally) that the Willis beam exhibits exotic properties, such as nonreciprocal behavior, perfect absorption and beam steering over a wide range of frequencies. The final results are new in the topical field of Willis metamaterials, and generally interesting. From this perspective, I find them suitable for publication in *Nat. Commun.* However, I have several issues with the manuscript and some of the statements therein, which should be addressed prior to its acceptance.

(A) The manuscript is hard to follow (at least for me), partially since necessary information is either assumed as common knowledge (an important example is given in the next point) or relegated to the supplementary information (SI). To my understanding, the manuscript should be able to stand on its own, where the SI should not be essential for the understanding of the main results. However, some of the main results appear only in the SI. For example, in the summary of the results appearing in the conclusion section, the authors list their development of a source-driven homogenization theory—this development appears only in the SI. Similarly, beam steering is described in the body of the text, however the readers are directed to SI for corresponding figures.

(B) The concept of bianisotropic polarizability tensor has been recently introduced in [1] for acoustics, and used thereafter by others in this context (acoustics), e.g., [2]. To my knowledge, the introduction in this manuscript of a polarization tensor for flexural waves is new. If it is not, please add references for more clarification. If it is indeed new, there is a need to clarify more its physical meaning and connection to the constitutive equations in the manuscript. Specifically, to me it is not clear how the authors associate some of the quantities and monopole (in-phase) and some as dipole (out-of-phase) quantities, e.g., how is the curvature associated with out-of-phase propagation? Perhaps an illustration as in [1] will be useful. The connection between the polarizability tensor and the constitutive properties of beams in Willis form also needs clarification. (Again, there is some explanation in the SI, but I would expect it to appear in a clearer and more concise form in the manuscript itself.)

(C) The authors refer to the symmetry property of the Willis coupling in the abstract and the body of the text. I find this statement as ambiguous: initially I understood it as a symmetry with respect to the components of the Willis coupling, but in the text it turns out it refers to the loss of reciprocity [3], or, equivalently, the loss of self-adjointness of effective constitutive operator [4]. This should be clarified.

(D) The authors claim that “It is also important to mention that the scattering properties of a Willis scatterer cannot be properly captured using simple the Euler beam theory because it lacks the degree of freedom associated shearing motion within the beam that is required to capture monopole-dipole coupling that is the finger-print of the Willis material response”. Why an Euler beam whose bending moment depends (not only on the curvature but also) on the transverse velocity, and whose linear momentum depends (not only on the transverse velocity but also) on the curvature [5] is not considered a beam with a Willis response?

(E) Some of the statements made in the text should be toned down. For example, in the abstract

the authors claim that "Willis coupling coefficients in Willis solids designed so far are symmetric". However, there are several works who already broke reciprocity in Willis materials theoretically [6] and experimentally [7] (I think that the last author also has a relevant theoretical work on the topic [8]).

Another statement that I find inexact is that "In summary, we have provided a theoretic and experimental study on an active Willis meta-layer in elastic plates...". The theoretical analysis is completely a one-dimensional analysis, and therefore is not applicable to plates, which are governed by equations of two spatial coordinates. In the SI, the authors themselves say that "Note that the approach demonstrated below can be easily extended to a Willis plate through the use of Mindlin-Reissner plate theory", i.e., they did not execute this extension. I suggest that the authors would either extend their approach as they say, or rephrase their statements.

(F) Finally, the manuscript should undergo a thorough proofreading, as it contains several typos (e.g., line 68 is missing the word "to" after "ability", in line 181 the words "is" and "leads" should be "are" and "lead" since they refer to the word "functions").

References

- [1] C. F. Sieck, A. Alù, and M. R. Haberman, Phys. Rev. B 96, 104303 (2017). [2] X. Su and A. N. Norris, Phys. Rev. B 98, 174305 (2018)
- [3] M. B. Muhlestein, C. F. Sieck, A. Alù, and M. R. Haberman, Proc. R. Soc. A 472, 20160604 (2016)
- [4] J. R. Willis, Proc. R. Soc. A 467, 1865 (2011)
- [5] R. P. Salomon, G. Shmuel, J. Mech. Phys. Solids 119 (2018)
- [6] L. Quan, D. L. Sounas, and A. Alu, Phys. Rev. Lett., 123, 064301(2019)
- [7] Y. Zhai, H.-S. Kwon, and B.-I. Popa, Phys. Rev. B 99, 220301(R) (2019)
- [8] H. Nassar, X. Xu, A. Norris, and G. Huang, J. Mech. Phys. Solids 101, 10 (2017)

Response to reviewer's comments (#1)

The authors would like to take this opportunity to thank the reviewer for his/her insights and suggestions, and have responded to the comments below. The paper has been modified accordingly.

This manuscript presents an active elastic wave metamaterial, based around a sensor which measures the symmetric component of the incident field, and is then able to generate controlled levels of both symmetric and anti-symmetric waves. This is presented as an example of non-reciprocal Willis coupling in a metamaterial, and this description is quite sound.

However, the authors model this system with what they call a "polarizability" model, attributing the symmetric radiation to excitation of a monopole mode, and anti-symmetric radiation to a dipole mode. This terminology does not seem well justified, for example the bending moment M is a higher order moment of the stress tensor, so describing it as a monopolar term is quite confusing. We should expect the degree of a multipole to correspond to some order of near-field spatial variation, or to a far-field radiation pattern.

Response: We thank the reviewer for their insight in regards to our interpretation of the physical behavior of the system under study. We agree with the reviewer that the mechanical "polarizability" may not have been properly introduced and that several intermediate steps were omitted in the original submission. These points are addressed in the updated manuscript (Pages 9-13) and described below.

We consider the mechanical active meta-layer as a point scatterer, and formulate its polarizability tensor, β , for flexural waves as

$$\tilde{\mathbf{Q}} = \beta \mathbf{F}_{\text{loc}}, \quad (\text{R1})$$

where $\mathbf{F}_{loc} = [\psi_{loc} \quad w_{loc} \quad F_{loc} \quad M_{loc}]^T$ is the incident wave field vector at the scatterer location, with ψ_{loc} , w_{loc} , F_{loc} and M_{loc} denoting local rotational angle, transverse displacement, shear force and bending moment, respectively. The vector $\tilde{\mathbf{Q}} = [\tilde{q}_0 \quad \tilde{f}_0 \quad \tilde{s}_0 \quad \tilde{p}_0]^T$ is the multipole vector, representing the “excited” (scattered) field caused by the interaction of the point scatterer with the local field, with \tilde{q}_0 , \tilde{f}_0 , \tilde{s}_0 and \tilde{p}_0 representing the body torque, transverse body force, shear strain and bending curvature, respectively. The definition provided in Eq. (R1) describes general scattering behavior of flexural waves incident on an arbitrary point scatterer, which is analogous to definitions of electromagnetic and acoustic polarizability tensors. The physical representations of the multipole vector and associated radiation patterns in a host beam are shown in Fig. R1. It can be seen that torque and shear strain generate antisymmetric modes, where outward propagating waves are out-of-phase when traveling in opposing directions, while transverse force and bending curvature generate symmetric modes, where outward propagating waves are in-phase when traveling in opposing directions.

Figure R1. Schematic of radiation patterns in the beam caused by the multipole vector $\tilde{\mathbf{Q}}$: (a) \tilde{q}_0 ; (b) \tilde{f}_0 ; (c) \tilde{s}_0 ; (d) \tilde{p}_0 .

To determine orders of those multipole components of $\tilde{\mathbf{Q}} = [\tilde{q}_0 \quad \tilde{f}_0 \quad \tilde{s}_0 \quad \tilde{p}_0]^T$, we conduct numerical simulations on a plate with excitations of body torque, body transverse force, shear strain and bending curvature applied in the middle portion of this plate. The out-of-plane displacement fields are shown in Fig. R2.

Figure R2. Simulated out-of-plane displacement field on a plate with excitations of body torque (a), body transverse force (b), bending curvature (c) and shear strain (d).

As shown in Fig. R2, it is clear that \tilde{f}_0 is a monopole quantity of order zero in the multipole expansion and that \tilde{q}_0 and \tilde{s}_0 are dipole quantities of order one. Similarly, ψ_{loc} , w_{loc} and F_{loc} are dipole, monopole and dipole quantities, respectively. Furthermore, as shown in Fig. R2c, the field associated with localized curvature, \tilde{p}_0 , results in a longitudinal quadrupole quantity. We thus justify \tilde{p}_0 and M_{loc} as longitudinal quadrupoles instead of monopoles. We modified terminologies of these multipoles in the revised manuscript.

Near-field and far-field radiation patterns due to these multipoles are discussed in details below because they are also related to the next question to be answered.

Furthermore, the authors first introduce a 4x4 matrix description, with 2 "dipole" and 2 "monopole" quantities in both source and induced quantities, but quickly reduce to only

considering a 2x2 matrix, without any real justification. It is quite possible that the other quantities do play a role in the experiment, but given the number of observable transmission/reflection coefficients, it is not possible to resolve all 16 coefficients. It would be much to simplify the model to describe symmetric and anti-symmetric components, without making claims about multipole orders, or unnecessarily trying to differentiate indistinguishable scattering mechanisms.

Response: As the reviewer pointed out, it is true that we can only retrieve the polarizability tensor in a 2×2 matrix by just using far-field radiation (propagating waves) displayed by wave transmissions and reflections and ignoring near-field variations (evanescent waves).

To obtain all the 16 effective polarizability coefficients of the meta-layer in the 4×4 matrix, we formulate the polarizability-retrieval method (with Timoshenko beam assumption) step by step by considering both propagated (far-field) and evanescent (near-field) wave solutions.

First, kinematic equations of the Timoshenko beam with external deformation sources can be written as:

$$\begin{aligned}\frac{\partial \psi}{\partial x} &= \kappa + p, \\ \frac{\partial w}{\partial x} - \psi &= \gamma + s,\end{aligned}\tag{R2}$$

where w , ψ , κ and γ respectively denote the transverse displacement, rotational angle, passive bending curvature and passive shear strain, while p and s respectively represent externally applied bending curvature and shear strain.

Second, conservation of the translational and rotational momentums of the Timoshenko beam with external loadings leads to the equations:

$$\begin{aligned}\rho_0 \frac{\partial^2 w}{\partial t^2} - \frac{\partial F}{\partial x} &= f, \\ J_0 \frac{\partial^2 \psi}{\partial t^2} + \frac{\partial M}{\partial x} - F &= q,\end{aligned}\tag{R3}$$

where M , F , f , q , ρ_0 and J_0 respectively denote the bending moment, shear force, externally applied transverse body force and body torque, mass density, and rotational inertia per unit length of the host beam.

Thirdly, the constitutive relations of this host beam are expressed as:

$$\begin{aligned}M &= -D_0 \kappa, \\ F &= AG\chi\gamma = G_0 \gamma,\end{aligned}\tag{R4}$$

where D_0 , G , A and χ respectively denote the bending stiffness, shear modulus, area of the cross section and Timoshenko coefficient.

Combining Eqs. (R2) - (R4), the governing equation of the Timoshenko beam with external force/torque loadings and deformation sources can be obtained as

$$\begin{aligned}D_0 \frac{\partial^4 w}{\partial x^4} + \left(\frac{\rho_0 \omega^2 D_0}{G_0} + J_0 \omega^2 \right) \frac{\partial^2 w}{\partial x^2} + \frac{(J_0 \omega^2 - G_0) \rho_0 \omega^2}{G_0} w \\ = \left(-\frac{J_0 \omega^2}{G_0} + 1 \right) f - \frac{D_0}{G_0} \frac{\partial^2 f}{\partial x^2} - \frac{\partial q}{\partial x} + D_0 \frac{\partial^2 p}{\partial x^2} + J_0 \omega^2 \frac{\partial s}{\partial x} + D_0 \frac{\partial^3 s}{\partial x^3},\end{aligned}\tag{R5}$$

where the time harmonic term, $e^{i\omega t}$, is dropped from Eq. (R5).

Considering a scatterer located at $x = 0$ in the Timoshenko beam. Its scattered waves due to the inhomogeneity can be equivalently regarded as “excited” (scattered) waves caused by a multipole vector, $\tilde{\mathbf{Q}}$, with external force/torque loadings and deformation sources in the Timoshenko beam being $f = \tilde{f}_0 \delta(x)$, $q = \tilde{q}_0 \delta(x)$, $p = \tilde{p}_0 \delta(x)$ and $s = \tilde{s}_0 \delta(x)$. Those excited wave fields can be expressed in terms of Green’s functions as:

$$w_s = \tilde{f}_0 G_f(x, 0) + \tilde{q}_0 G_q(x, 0) + \tilde{p}_0 G_p(x, 0) + \tilde{s}_0 G_s(x, 0), \quad (\text{R6})$$

where

$$\begin{aligned} G_f(x, 0) &= A_1 e^{-ik_1|x|} + a_1 A_1 e^{-k_2|x|}, \\ G_q(x, 0) &= \text{sgn}(x) \left(A_2 e^{-ik_1|x|} + a_2 A_2 e^{-k_2|x|} \right), \\ G_p(x, 0) &= A_3 e^{-ik_1|x|} + a_3 A_3 e^{-k_2|x|}, \\ G_s(x, 0) &= \text{sgn}(x) \left(A_4 e^{-ik_1|x|} + a_4 A_4 e^{-k_2|x|} \right), \end{aligned}$$

with

$$\begin{aligned} k_1 &= -i \sqrt{\frac{-\alpha - \sqrt{\alpha^2 - 4D_0\beta}}{2D_0}}, \quad k_2 = \sqrt{\frac{-\alpha + \sqrt{\alpha^2 - 4D_0\beta}}{2D_0}}, \quad \alpha = \frac{\rho_0 \omega^2 D_0}{G_0} + J_0 \omega^2, \\ \beta &= \frac{(J_0 \omega^2 - G_0) \rho_0 \omega^2}{G_0}, \quad A_1 = \frac{1}{2(ik_1 + a_1 k_2) G_0}, \quad a_1 = -\frac{(-\rho_0 \omega^2 + G_0 k_1^2) k_2}{(\rho_0 \omega^2 + G_0 k_2^2) k_1} i, \quad A_2 = \frac{1}{2(ik_1 + c_0 k_2) D_0 b_0}, \\ c_0 &= \frac{i(\rho_0 \omega^2 + G_0 k_2^2) k_1}{(-\rho_0 \omega^2 + G_0 k_1^2) k_2}, \quad b_0 = \frac{(-\rho_0 \omega^2 + G_0 k_1^2)}{i G_0 k_1}, \quad a_2 = -1, \quad A_3 = \frac{1}{2 \left(b_0 - b_3 \frac{b_0 + ik_1}{b_3 + k_2} \right)}, \\ b_3 &= \frac{-\rho_0 \omega^2 - G_0 k_2^2}{G_0 k_2}, \quad a_3 = -\frac{b_0 + ik_1}{b_3 + k_2}, \quad A_4 = \frac{1}{2(1 + d_0)}, \quad d_0 = \frac{-\rho_0 \omega^2 + G_0 k_1^2}{\rho_0 \omega^2 + G_0 k_2^2} \text{ and } a_4 = 1. \end{aligned}$$

The Green's functions satisfy the following:

$$\begin{aligned}
D_0 \frac{\partial^4 G_f(x,0)}{\partial x^4} + \left(\frac{\rho_0 \omega^2 D_0}{G_0} + J_0 \omega^2 \right) \frac{\partial^2 G_f(x,0)}{\partial x^2} + \frac{(J_0 \omega^2 - G_0) \rho_0 \omega^2}{G_0} G_f(x,0) \\
= \left(-\frac{J_0 \omega^2}{G_0} + 1 \right) \tilde{f}_0 \delta(x) - \frac{D_0}{G_0} \frac{\partial^2 [\tilde{f}_0 \delta(x)]}{\partial x^2}, \\
D_0 \frac{\partial^4 G_q(x,0)}{\partial x^4} + \left(\frac{\rho_0 \omega^2 D_0}{G_0} + J_0 \omega^2 \right) \frac{\partial^2 G_q(x,0)}{\partial x^2} + \frac{(J_0 \omega^2 - G_0) \rho_0 \omega^2}{G_0} G_q(x,0) \\
= -\frac{\partial [\tilde{q}_0 \delta(x)]}{\partial x}, \\
D_0 \frac{\partial^4 G_p(x,0)}{\partial x^4} + \left(\frac{\rho_0 \omega^2 D_0}{G_0} + J_0 \omega^2 \right) \frac{\partial^2 G_p(x,0)}{\partial x^2} + \frac{(J_0 \omega^2 - G_0) \rho_0 \omega^2}{G_0} G_p(x,0) \\
= D_0 \frac{\partial^2 [\tilde{p}_0 \delta(x)]}{\partial x^2}, \\
D_0 \frac{\partial^4 G_s(x,0)}{\partial x^4} + \left(\frac{\rho_0 \omega^2 D_0}{G_0} + J_0 \omega^2 \right) \frac{\partial^2 G_s(x,0)}{\partial x^2} + \frac{(J_0 \omega^2 - G_0) \rho_0 \omega^2}{G_0} G_s(x,0) \\
= J_0 \omega^2 \frac{\partial [\tilde{s}_0 \delta(x)]}{\partial x} + D_0 \frac{\partial^3 [\tilde{s}_0 \delta(x)]}{\partial x^3}. \tag{R7}
\end{aligned}$$

On the other hand, when the active meta-layer is excited with an incident wave, the scattered wave fields from the meta-layer (considered as a scatterer) can also be numerically extracted in the form of (Fig. R3)

$$\begin{aligned}
w_s^+ &= B_1 e^{-ik_1 x} + B_2 e^{-k_2 x}, \\
w_s^- &= B_3 e^{ik_1 x} + B_4 e^{k_2 x}, \tag{R8}
\end{aligned}$$

where B_1 and B_3 denote wave amplitudes of left- and right-propagating (far-field) waves, and B_2 and B_4 represent wave amplitude of left- and right- evanescent (near field) waves.

Figure R3. Illustrations of coefficients of scattered wave fields.

Therefore, the excited multipole vector, $\tilde{\mathbf{Q}}$, of the meta-layer (scatterer) can be numerically determined by equalizing the excited wave fields in Eq. (R6) and the scattered wave fields in Eq. (R8) as

$$\begin{bmatrix} \tilde{f}_0 \\ \tilde{q}_0 \\ \tilde{p}_0 \\ \tilde{s}_0 \end{bmatrix} = \begin{bmatrix} A_1 & A_2 & A_3 & A_4 \\ A_1 a_1 & A_2 a_2 & A_3 a_3 & A_4 a_4 \\ A_1 & -A_2 & A_3 & -A_4 \\ A_1 a_1 & -A_2 a_2 & A_3 a_3 & -A_4 a_4 \end{bmatrix}^{-1} \begin{bmatrix} B_1 \\ B_2 \\ B_3 \\ B_4 \end{bmatrix}. \quad (\text{R9})$$

To retrieve the effective 4×4 polarizability tensor of the meta-layer attached with piezoelectric patches, four independent numerical tests are conducted. In the first two numerical tests ($j = 1, 2$), the transverse force source is located in the left or right ends of the host beam such that only propagating (far-field) waves can be measured by the active meta-layer (see Cases I and II in Fig. R4). In the last two numerical tests ($j = 3, 4$), the transverse force source is located adjacent to the left and right sides of the meta-layer such that both propagating (far-field) and evanescent (near-field) waves can be measured by the active meta-layer (see Cases III and IV in Fig. R4).

Figure R4. Illustration of numerical tests under the four transverse forces at different locations

For the j -th test, the excited multipole vector of the active meta-layer,

$\tilde{\mathbf{Q}}^{(j)} = [\tilde{q}_0^{(j)} \quad \tilde{f}_0^{(j)} \quad \tilde{s}_0^{(j)} \quad \tilde{p}_0^{(j)}]^T$, is firstly determined based on Eq. (R9), when the control is on.

The local field vector at the active meta-layer location, $\mathbf{F}_{\text{loc}}^{(j)} = [\psi_{\text{loc}}^{(j)} \quad w_{\text{loc}}^{(j)} \quad F_{\text{loc}}^{(j)} \quad M_{\text{loc}}^{(j)}]^T$, is

then measured, when the control is off. Combining excited multipole vectors with their corresponding local field vectors for the four cases ($j = 1, 2, 3, 4$), one can retrieve the 16 polarizability coefficients, according to Eq. (R1), as

$$\begin{bmatrix} \beta_1 \\ \beta_2 \\ \beta_3 \\ \beta_4 \end{bmatrix} = \begin{bmatrix} \mathbf{T}_{\text{loc}}^{(11)} & \mathbf{T}_{\text{loc}}^{(12)} & \mathbf{T}_{\text{loc}}^{(13)} & \mathbf{T}_{\text{loc}}^{(14)} \\ \mathbf{T}_{\text{loc}}^{(21)} & \mathbf{T}_{\text{loc}}^{(22)} & \mathbf{T}_{\text{loc}}^{(23)} & \mathbf{T}_{\text{loc}}^{(24)} \\ \mathbf{T}_{\text{loc}}^{(31)} & \mathbf{T}_{\text{loc}}^{(32)} & \mathbf{T}_{\text{loc}}^{(33)} & \mathbf{T}_{\text{loc}}^{(34)} \\ \mathbf{T}_{\text{loc}}^{(41)} & \mathbf{T}_{\text{loc}}^{(42)} & \mathbf{T}_{\text{loc}}^{(43)} & \mathbf{T}_{\text{loc}}^{(44)} \end{bmatrix}^{-1} \begin{bmatrix} \tilde{\mathbf{Q}}^{(1)} \\ \tilde{\mathbf{Q}}^{(2)} \\ \tilde{\mathbf{Q}}^{(3)} \\ \tilde{\mathbf{Q}}^{(4)} \end{bmatrix}, \quad (\text{R10})$$

where

$$\beta_i = [\beta_{i1} \quad \beta_{i2} \quad \beta_{i3} \quad \beta_{i4}]^T, \quad i = 1, 2, 3, 4$$

$$\mathbf{T}_{\text{loc}}^{(i1)} = \begin{bmatrix} (\mathbf{F}_{\text{loc}}^{(i)})^T \\ \mathbf{0} \\ \mathbf{0} \\ \mathbf{0} \end{bmatrix}, \quad \mathbf{T}_{\text{loc}}^{(i2)} = \begin{bmatrix} \mathbf{0} \\ (\mathbf{F}_{\text{loc}}^{(i)})^T \\ \mathbf{0} \\ \mathbf{0} \end{bmatrix}, \quad \mathbf{T}_{\text{loc}}^{(i3)} = \begin{bmatrix} \mathbf{0} \\ \mathbf{0} \\ (\mathbf{F}_{\text{loc}}^{(i)})^T \\ \mathbf{0} \end{bmatrix}, \quad \mathbf{T}_{\text{loc}}^{(i4)} = \begin{bmatrix} \mathbf{0} \\ \mathbf{0} \\ \mathbf{0} \\ (\mathbf{F}_{\text{loc}}^{(i)})^T \end{bmatrix}, \quad i = 1, 2, 3, 4$$

To ensure that all terms in the tensor have the same units, we normalize polarizability tensors by firstly normalizing the left- and right-hand sides of Eq. (R1) to the strain and then multiplying k_1 to the whole tensor as shown:

$$\begin{bmatrix} \beta'_{11} & \beta'_{12} & \beta'_{13} & \beta'_{14} \\ \beta'_{21} & \beta'_{22} & \beta'_{23} & \beta'_{24} \\ \beta'_{31} & \beta'_{32} & \beta'_{33} & \beta'_{34} \\ \beta'_{41} & \beta'_{42} & \beta'_{43} & \beta'_{44} \end{bmatrix} = \begin{bmatrix} \beta_{11}/G_0 & k_1\beta_{12}/G_0 & k_1\beta_{13} & k_1^2 D_0\beta_{14}/G_0 \\ \beta_{21}/(k_1G_0) & \beta_{22}/G_0 & \beta_{23} & k_1 D_0\beta_{24}/G_0 \\ \beta_{31} & k_1\beta_{32} & k_1\beta_{33}G_0 & k_1^2 D_0\beta_{34} \\ \beta_{41}/k_1 & \beta_{42} & \beta_{43}G_0 & k_1 D_0\beta_{44} \end{bmatrix}. \quad (\text{R11})$$

Figure R5 shows amplitudes of those retrieved normalized complex polarizabilities of the active meta-layers demonstrated in Fig. 3d. We find that $|\beta'_{34}|$ and $|\beta'_{44}|$ are close to 2 and other polarizabilities are near zero. The form of the retrieved polarizability tensor agrees with the analysis in the original manuscript. Furthermore, by implementing an active effective scatterer (characterized by retrieved polarizabilities mentioned above) into a host beam (Fig. R6), the retrieved polarizabilities are validated through comparisons of the wave fields from numerical predictions. It is clearly seen that the wave fields calculated are almost the same with those in Figs. 3c and 3d where the meta-layer is physically applied. Based on this analysis and the numerical observations, we can conclude that other quantities (those near zero polarizabilities) in the polarizability tensor of the active meta-layer don't play any roles in the experiment.

The discussion has been added in the revised manuscript and Supplementary Information.

Figure R5. Amplitude of those retrieved polarizabilities.

Figure R6. Numerically simulated wave fields at 10 kHz using the effective model characterized by the polarizabilities in Fig. R5.

Overall this theoretical approach is quite difficult to follow, and seems unnecessary to describe the experiment. For example, the required voltage transfer functions given by Eq (24) in the supplementary material do not require the polarizability model, but just depend on simple quantities such as required transmission, separation between transducers etc. In particular, the source driven homogenization does not seem to relate to the experimental part, or to make any new predictions, and homogenization is not meaningful for the considered cases of a single scatterer and an inhomogeneous line array of scatterers.

Response: Thank you for this critical suggestion. It is true that voltage transfer functions can be given in terms of wave transmissions and reflections along with other physical parameters by considering electromechanical coupling calculations (Supplementary Note: Derivation of transfer functions H_1 and H_2). However, it is difficult to gain mechanical insights of the design from straightforward calculations due to the lack of direct mechanical representations. However, the polarizability model (illustrating general relations between local fields and excited multipoles) provides a simple and efficient way to characterize effective mechanical responses of the active meta-layer and is one of the key contributions of the research presented in this work. In particular, in the design of the active meta-layer, the symmetric part of actuating voltages ($V_+ = (V_{a1} + V_{a2})/2$) produces an effective bending curvature, and the antisymmetric part of actuating

voltages ($V_s = (V_{a1} - V_{a2})/2$) induces an effective shear strain. Because the sensing voltage V_s is proportional to the local bending curvature caused by incident waves, electrical transfer functions and mechanical polarizabilities are intrinsically correlated to describe this coupling. According to definitions of polarizabilities, the relations between electric transfer functions and mechanical polarizabilities can be obtained as $\beta_{44} = \frac{\chi_q (H_1 + H_2)}{2}$ and $\beta_{34} = \frac{\chi_f (H_1 - H_2)}{2}$,

where χ_q and χ_f respectively denote factors describing the electromechanical coupling of symmetric and antisymmetric modes (this relation is indeed presented in original version). Furthermore, the polarizability model should bring a fresh understanding and provide a precise design principle of the active meta-layer for arbitrary wave control. From a point of view in polarizabilities, a non-zero symmetric-to-antisymmetric coupling in the meta-layer is responsible for breaking reciprocity, and we can independently control transmission and reflection for the incidence from one direction by using active meta-layer if its polarizabilities have non-zero symmetric-to-symmetric and symmetric-to-antisymmetric couplings. The polarizability model may also be used to guide new designs of other types of active meta-layers by selecting other coupling components to be nonzero. In the revised version, we further articulate the importance of polarizability model and discuss its connection to the multiphysics calculations.

The purpose of source-driven homogenization model is to determine effective material parameters of a Willis beam with a periodic array of the active meta-layers (scatterers). It is an important step to extend and generalize the developed Willis meta-layer to a Willis material for future unprecedented wave control. As the reviewer pointed out, the manuscript does not contain further experimental or numerical studies on the Willis beam with an array of active scatterers. Therefore, in the revised manuscript, we will only mention the main results and significance of

the source-driven homogenization model in the last section, and move the source-driven homogenization model into Supplementary Information for detailed explanations.

On the other hand, the experimental aspects of this paper seem novel, well conducted, and quite interesting. In my view, this part of the paper clearly meets the level of novelty I would expect of Nature Communications, and could readily be understood by abroad audience. If the experimental results were given more prominence, and the theoretical part was reduced to the minimum necessary to interpret the experiments, this would be quite a solid paper.

Response: We highly appreciate the positive comments regarding the experimental part. According to the reviewer's suggestions, we reorganized the manuscript. In the revised version, we start with experiments, then transition to the polarizability model to characterize effective scattering properties of the active meta-layer, and finally summarize general rules in flexural wave control. Furthermore, we only mention the important conclusions from the source-driven homogenization model in the last section and move the section of the source-driven homogenization model into Supplementary Information. We believe the theoretical part has been reduced to the minimum necessary to interpret the experiments in the main text and that the experimental results have been given more prominence with these revisions.

I note a few minor issues:

- In several places numerical simulations are performed, however it is never described how these are done. It appears that some form of full wave elastic and piezoelectric coupled model is used.

Response: The descriptions of numerical simulations are added into the Methods section, as shown below:

“3D numerical simulations of harmonic wave propagations through the Willis meta-layer are conducted using the commercial finite element software, COMSOL Multiphysics. In numerical simulations, the 3D linear piezoelectric constitutive law is applied to the piezoelectric patches. The sensing signal is obtained by integrating free charges over a surface of an electrode of the piezoelectric sensor. The two electrodes on the piezoelectric sensor have zero electric potential. The incident flexural wave is generated by applying a harmonic voltage across a piezoelectric patch bonded to the host beam on the left- and right-hand sides of the meta-layer. Two perfectly matched layers (PMLs) are attached to both ends of the host beam in order to suppress reflected waves from the boundaries. Two displacement probes are defined on the host beam in the right- and left- hand sides of the meta-layer to measure the out-of-plane displacement and calculate wave transmission and reflection coefficients.”

- On line 38 the statement "furnish unsurpassed control" seems like hype, and should be replaced with something more substantive.

Response: In the revise manuscript, the sentence has been changed to “The stress-velocity and momentum-strain coupling offered by Willis materials provides appealing solutions in many applications including but not limited to perfect elastic wave cloaking [25], independent control of transmissions and reflections [17], perfect wavefront manipulations [19] and reciprocity breaking [12,16].”.

- On lines 49-50, vague mention is made of the constraints of passive media for certain applications. This claim should be made more concrete and backed up by examples.

Response: We modified this sentence accordingly as

“In addition, the coupling constants are intrinsically connected, posing fundamental constraints on applications for Willis materials such as broadband operation, violation of reciprocal propagation, and independent and non-conservative control of wave transmission and reflection.”

- On lines 193-194, it is stated that the scattering in the passive state is negligible. This statement should be quantified.

Response: According to the reviewer’s comments, we calculate transmittance of the meta-layer without electrical control at frequencies of interested using numerical simulations (Fig. R7). It is found transmittance is very close to 1 in the presence of the passive meta-layer. Therefore, scattering in the passive state is negligible. We add this figure into the SI.

Figure R7. Numerically calculated transmittance of the meta-layer without electrical control at frequencies from 9 to 18 kHz.

- On line 266, reference titles suggest that reference [21] should have been cited instead of [20]

Response: All the reference numbers have been corrected in the revised manuscript.

Response to reviewer's comments (#2)

The authors would like to take this opportunity to thank the reviewer for his/her insights and suggestions, and have responded to the comments below. The paper has been modified accordingly.

The broad context of the work is Willis materials—a class of mechanical metamaterials with unusual dynamic constitutive relations, namely, velocity-dependent stress and strain-dependent linear momentum. The authors propose and experimentally realize a structural counterpart of Willis materials, by considering beams undergoing flexural motions with analogous cross-coupling. To my knowledge, the novelty in this work is the implementation of active elements in the settings of Willis beams. As a result of the integration of these elements, the authors demonstrate (theoretically, numerically, and experimentally) that the Willis beam exhibits exotic properties, such as nonreciprocal behavior, perfect absorption and beam steering over a wide range of frequencies. The final results are new in the topical field of Willis metamaterials, and generally interesting. From this perspective, I find them suitable for publication in Nat. Commun. However, I have several issues with the manuscript and some of the statements therein, which should be addressed prior to its acceptance.

Response: We thank the reviewer for overall positive comments.

(A) The manuscript is hard to follow (at least for me), partially since necessary information is either assumed as common knowledge (an important example is given in the next point) or relegated to the supplementary information (SI). To my understanding, the manuscript should be able to stand on its own, where the SI should not be essential for the understanding of the main results. However, some of the main results appear only in the SI. For example, in the summary of

the results appearing in the conclusion section, the authors list their development of a source-driven homogenization theory—this development appears only in the SI. Similarly, beam steering is described in the body of the text, however the readers are directed to SI for corresponding figures.

Response: To address the reviewer’s concerns, we have reorganized the manuscript. In the revised version, we start with experimental results, and then introduce the polarizability model to characterize effective scattering properties of the active meta-layer and summarize general rules in flexural wave control. We have re-written the polarizability model (see the response that follows) and further articulate the importance of polarizability model.

In the main text, we concentrate on the meta-layer design, its experimental and numerical results and effective material parameters in the form of polarizability tensor, which would be aligned with the comments raised by the first reviewer. Whereas, the source-driven homogenization model proposed is indeed an important step to extend and generalize the developed Willis meta-layer to a Willis material, which we feel is a relevant and important contribution. Therefore, we mention its importance and conclusion in the last section, and move the source-driven homogenization model into Supplementary Information. We hope this modification also fits the comments raised by the reviewer.

Finally, we moved figures showing wave absorption and steering from the SI to the main text to more clearly demonstrate the degree of wave control afforded by the meta-layer in on a plate. We now believe that all of the most important and necessary information is included in the main text, and that the manuscript stands on its own after this change.

(B) The concept of bianisotropic polarizability tensor has been recently introduced in [1] for

acoustics, and used thereafter by others in this context (acoustics), e.g., [2]. To my knowledge, the introduction in this manuscript of a polarization tensor for flexural waves is new. If it is not, please add references for more clarification. If it is indeed new, there is a need to clarify more its physical meaning and connection to the constitutive equations in the manuscript. (1) Specifically, to me it is not clear how the authors associate some of the quantities and monopole (in-phase) and some as dipole (out-of-phase) quantities, e.g., how is the curvature associated with out-of-phase propagation? Perhaps an illustration as in [1] will be useful. (2) The connection between the polarizability tensor and the constitutive properties of beams in Willis form also needs clarification. (Again, there is some explanation in the SI, but I would expect it to appear in a clearer and more concise form in the manuscript itself.)

Response: We first (1) interpret the physical meaning of the polarization tensor and then (2) illustrate the connection between the polarizability tensor and the constitutive properties of beams in the Willis form.

(1) Using “polarizabilities” to characterize elastic scatterers for flexural waves is new to the best of authors’ knowledge. We have added more details regarding its physical meaning in the manuscript as the followings:

We consider the mechanical active meta-layer as a point scatterer, and formulate its polarizability tensor, β , for flexural waves as

$$\tilde{\mathbf{Q}} = \beta \mathbf{F}_{loc}, \quad (\text{R1})$$

where $\mathbf{F}_{loc} = [\psi_{loc} \quad w_{loc} \quad F_{loc} \quad M_{loc}]^T$ is the incident field vector at the scatterer location, and ψ_{loc} , w_{loc} , F_{loc} and M_{loc} respectively denote local rotational angle, transverse displacement, shear force and bending moment. $\tilde{\mathbf{Q}} = [\tilde{q}_0 \quad \tilde{f}_0 \quad \tilde{s}_0 \quad \tilde{p}_0]^T$ is the multipole vector, representing

the “excited” (scattered) waves caused by the point scatterer. Specifically, \tilde{q}_0 , \tilde{f}_0 , \tilde{s}_0 and \tilde{p}_0 represent the body torque, transverse body force, shear strain and bending curvature, respectively. The definition provided in Eq. (R1) describes general scattering behavior of flexural waves incident on an arbitrary point scatterer, which is analogous to definitions of electromagnetic and acoustic polarizability tensors. The physical representations of the multipole vector and associated radiation patterns in a host beam are shown in Fig. R1. It can be seen that torque and shear strain generate antisymmetric modes, where outward propagating waves are out-of-phase when traveling in opposing directions, while transverse force and bending curvature generate symmetric modes, where outward propagating waves are in-phase when traveling in opposing directions.

Figure R1. Schematic of radiation patterns in the beam caused by the multipole vector $\tilde{\mathbf{Q}}$: (a) \tilde{q}_0 ; (b) \tilde{f}_0 ; (c) \tilde{s}_0 ; (d) \tilde{p}_0 .

To determine orders of those multipole components of $\tilde{\mathbf{Q}} = [\tilde{q}_0 \quad \tilde{f}_0 \quad \tilde{s}_0 \quad \tilde{p}_0]^T$, we conduct numerical simulations on a plate with excitations of body torque, body transverse force, shear strain and bending curvature applied in the middle portion of this plate. The out-of-plane displacement fields are shown in Fig. R2.

Figure R2. Simulated out-of-plane displacement field on a plate with excitations of body torque (a), body transverse force (b), bending curvature (c) and shear strain (d).

As shown in Fig. R2, it is clear that \tilde{f}_0 is a monopole quantity of order zero in the multipole expansion and that \tilde{q}_0 and \tilde{s}_0 are dipole quantities of order one. Similarly, ψ_{loc} , w_{loc} and F_{loc} are dipole, monopole and dipole quantities, respectively. Furthermore, as shown in Fig. R2c, the field associated with localized curvature, \tilde{p}_0 , results in a longitudinal quadrupole quantity. We thus justify \tilde{p}_0 and M_{loc} as longitudinal quadrupoles instead of monopoles. We modified terminologies of these multipoles in the revised manuscript.

According to the definition of the polarizability tensor, its diagonal terms can be rewritten as

$$\beta_{11} = i\omega Z_{eff}^{(r)}, \quad \beta_{22} = i\omega Z_{eff}^{(t)}, \quad \beta_{33} = \frac{1}{G_{eff}} \quad \text{and} \quad \beta_{44} = \frac{1}{D_{eff}},$$

where $Z_{eff}^{(r)}$, $Z_{eff}^{(t)}$, G_{eff} and D_{eff} denote the effective impedance of rotational motion, effective impedance of transverse motion, effective shear stiffness and effective bending stiffness, respectively. These are physical meanings for diagonal terms. On the other hand, the off-diagonal terms represent all possible cross-couplings between local fields and excited multipoles, i.e. β_{13} and β_{31} represent antisymmetric-antisymmetric coupling between two dipole quantities, β_{24} and β_{42} represent symmetric-

symmetric coupling between monopole and multipole quantities, and β_{12} , β_{21} , β_{23} , β_{32} , β_{34} , β_{43} , β_{14} and β_{41} represent symmetric-antisymmetric couplings.

(2) For completeness, we add in this response below all the details of the connection between the polarizability tensor and constitutive properties of Willis beams, which are also related to Questions (C) and (D).

To clearly show this connection, we first define material parameters of Willis beams based on constitutive relations of a 3D Willis medium characterized by

$$\begin{aligned}\sigma_{ij} &= c_{ijkl}u_{k,l} + S_{ijk}v_k, \\ \mu_i &= \tilde{S}_{ijk}u_{j,k} + \rho_{ij}v_j,\end{aligned}\tag{R2}$$

where σ_{ij} , μ_i , u_k and v_k represent stress, momentum, displacement and velocity of the Willis medium, respectively, c_{ijkl} and ρ_{ij} are elastic and dynamic mass density tensors, and S_{ijk} and \tilde{S}_{ijk} denote Willis coupling tensors. Considering plane stress assumptions and assuming $\sigma_{33} = 0$ due to traction-free boundary conditions of the beam, Equation (R2) can be reduced as

$$\begin{aligned}\begin{bmatrix} \sigma_1 \\ \sigma_5 \end{bmatrix} &= \begin{bmatrix} \bar{c}_{11} & \bar{c}_{15} \\ \bar{c}_{51} & \bar{c}_{55} \end{bmatrix} \begin{bmatrix} \varepsilon_1 \\ \varepsilon_5 \end{bmatrix} + \begin{bmatrix} \bar{S}_{11} & \bar{S}_{13} \\ \bar{S}_{51} & \bar{S}_{53} \end{bmatrix} \begin{bmatrix} v_1 \\ v_3 \end{bmatrix}, \\ \begin{bmatrix} \mu_1 \\ \mu_3 \end{bmatrix} &= \begin{bmatrix} \hat{S}_{11} & \hat{S}_{15} \\ \hat{S}_{31} & \hat{S}_{35} \end{bmatrix} \begin{bmatrix} \varepsilon_1 \\ \varepsilon_5 \end{bmatrix} + \begin{bmatrix} \rho & 0 \\ 0 & \rho \end{bmatrix} \begin{bmatrix} v_1 \\ v_3 \end{bmatrix}.\end{aligned}\tag{R3}$$

Note that Voigt notation is employed in Eq. (R3) for matrix expressions. Here, we apply Timoshenko beam assumptions $u_1 = -z\psi$ with ψ denoting the rotational angle of the beam section. The normal and shear strains are

$$\begin{aligned}\varepsilon_1 &= -z \frac{\partial \psi}{\partial x}, \\ \varepsilon_5 &= \frac{\partial u_3}{\partial x} - \psi.\end{aligned}\tag{R4}$$

Integrating the stresses and momenta through the thickness of the Willis beam lead to expressions for the bending moment, M , rotational momentum, J , shear force, F , and transverse momentum, μ , in terms of the material properties and geometry of the beam which are, respectively, expressed as

$$\begin{aligned}
M &= \int_{-\frac{h}{2}}^{\frac{h}{2}} z \sigma_1 dz = -D \frac{\partial \psi}{\partial x} - S_{11}^{(b)} \dot{\psi}, \\
J &= -\int_{-\frac{h}{2}}^{\frac{h}{2}} z \mu_1 dz = S_{11}^{(h)} \frac{\partial \psi}{\partial x} + I \dot{\psi}, \\
F &= \int_{-\frac{h}{2}}^{\frac{h}{2}} \sigma_5 dz = g \varepsilon_5 + S_{53}^{(b)} \dot{u}_3, \\
\mu &= \int_{-\frac{h}{2}}^{\frac{h}{2}} \mu_3 dz = S_{35}^{(h)} \varepsilon_5 + \rho_0 \dot{u}_3,
\end{aligned} \tag{R5}$$

where $D = \frac{h^3 \bar{c}_{11}}{12}$, $S_{11}^{(b)} = \frac{h^3 \bar{S}_{11}}{12}$, $S_{11}^{(h)} = \frac{h^3 \hat{S}_{11}}{12}$, $I = \frac{h^3 \rho}{12}$, $g = h \bar{c}_{55}$, $S_{53}^{(b)} = h \bar{S}_{53}$, $S_{35}^{(h)} = h \hat{S}_{35}$ and

$\rho_0 = h \rho$ is the density per unit area with h being the thickness of the beam. According to Eq.

(R5), the constitutive relations of the Willis beam can be rewritten in the following matrix form

as

$$\begin{bmatrix} \kappa \\ J \\ \gamma \\ \mu \end{bmatrix} = \begin{bmatrix} -\bar{D} & -\bar{S}_{11}^{(b)} & 0 & 0 \\ -\bar{S}_{11}^{(h)} & \bar{I} & 0 & 0 \\ 0 & 0 & \bar{g} & -\bar{S}_{53}^{(b)} \\ 0 & 0 & \bar{S}_{35}^{(h)} & \bar{\rho} \end{bmatrix} \begin{bmatrix} M \\ \dot{\psi} \\ F \\ V \end{bmatrix}. \tag{R6}$$

where $\kappa = \frac{\partial \psi}{\partial x}$ is the local curvature, $\gamma = \varepsilon_5$ is the local shear strain, $V = \dot{u}_3$ represents the

transverse velocity, $\bar{D} = \frac{1}{D}$ is the bending compliance, $\bar{g} = \frac{1}{g}$ is the inverse of the shear modulus

of the beam material, and the remaining terms depend on the effective Willis coupling

coefficients of the structured beam $\bar{S}_{11}^{(b)} = \frac{S_{11}^{(b)}}{D}$, $\bar{S}_{11}^{(h)} = \frac{S_{11}^{(h)}}{D}$, $\bar{I} = I - S_{11}^{(h)} \bar{S}_{11}^{(b)}$, $\bar{S}_{53}^{(b)} = \frac{S_{53}^{(b)}}{g}$,

$\bar{S}_{35}^{(h)} = \frac{S_{35}^{(h)}}{g}$ and $\bar{\rho} = \rho_0 - S_{35}^{(h)} \bar{S}_{53}^{(b)}$. From Eq. (R6), we find the Willis couplings in off-diagonal

terms of the constitutive matrix respectively represent couplings between bending moment and rotational velocity and between shear force and transverse velocity.

Next, we develop a source-driven homogenization theory to illustrate the connection between the polarizability tensor (Eq. (R1)) and constitutive properties of Willis beams, which is a result of subwavelength periodic mechanical meta-layers (scatterers) embedded into a background beam.

Consider Timoshenko beam assumptions, conservations of translational and rotational momentums and kinematic equations of a beam are

$$\begin{aligned} \frac{\partial \psi}{\partial x} &= \kappa + p, \\ \frac{\partial V}{\partial x} &= \dot{\gamma} + \dot{\psi} + \dot{s}, \\ \frac{\partial F}{\partial x} &= \dot{\mu} - f, \\ \frac{\partial M}{\partial x} &= -J + F + q, \end{aligned} \tag{R7}$$

where q, f, s and p are source terms that represent the externally applied body torque, transverse body force, shear strain and bending curvature, respectively. Assume source distribution terms with amplitudes p_{ext} , s_{ext} , f_{ext} and q_{ext} and time-harmonic term, $e^{i(kx + \omega t)}$, where k is the wavenumber along x -direction. Conservation relations in Eq. (R7) for a background beam with those source distributions (Fig R3a) can be written in the spectrum domain as

$$\begin{aligned}
ik\psi_{ext} &= -\bar{D}_0 M_{ext} + p_{ext}, \\
ikW_{ext} &= \bar{g}_0 F_{ext} + \psi_{ext} + s_{ext}, \\
ikF_{ext} &= -\omega^2 \bar{\rho}_0 W_{ext} - f_{ext}, \\
ikM_{ext} &= \omega^2 \bar{I}_0 \psi_{ext} + F_{ext} + q_{ext}.
\end{aligned} \tag{R8}$$

where the mass density of the background beam is $\bar{\rho}_0$, bending compliance is \bar{D}_0 , shear compliance is, \bar{g}_0 , and rotational inertia is \bar{I}_0 .

Figure R3. Conceptual illustration of the source-driven homogenization procedure: (a) External sources applied on the background beam; (b) External sources applied on the background beam with periodic scatterers; (c) External sources applied on the effective beam.

We then introduce a periodic array of scatterers into the background beam (Fig. R3b), conservations of transverse and angular momentums and the kinematic equations with the same source distributions in Eq. (R8) read

$$\begin{aligned}
\frac{\partial \psi(x)}{\partial x} &= -\bar{D}_0 M(x) + \tilde{p}(x) + p_{ext} e^{ikx}, \\
\frac{\partial W(x)}{\partial x} &= \bar{g}_0 F(x) + \psi(x) + \tilde{s}(x) + s_{ext} e^{ikx}, \\
\frac{\partial F(x)}{\partial x} &= -\omega^2 \bar{\rho}_0 W(x) - \tilde{f}(x) - f_{ext} e^{ikx}, \\
\frac{\partial M(x)}{\partial x} &= \omega^2 \bar{I}_0 \psi(x) + F(x) + \tilde{q}(x) + q_{ext} e^{ikx},
\end{aligned} \tag{R9}$$

where $\tilde{p}(x) = [-\bar{D}(x) + \bar{D}_0] M(x)$, $\tilde{s}(x) = [\bar{g}(x) - \bar{g}_0] F(x)$, $\tilde{f}(x) = [\omega^2 \bar{\rho}(x) - \omega^2 \bar{\rho}_0] W(x)$ and $\tilde{q}(x) = [\omega^2 \bar{I}(x) - \omega^2 \bar{I}_0] \psi(x)$, accounting for the contrast in bending compliance, shear compliance, mass density and rotational inertia, respectively, between the background beam and the inhomogeneities. For the continuous source distributions, the effective field amplitudes for a representative volume element can be uniquely determined by [1]

$$\begin{aligned}
ik\psi_{eff} &= -\bar{D}_0 M_{eff} + \tilde{p}_{eff} + p_{ext}, \\
ikW_{eff} &= \bar{g}_0 F_{eff} + \psi_{eff} + \tilde{s}_{eff} + s_{ext}, \\
ikF_{eff} &= -\omega^2 \bar{\rho}_0 W_{eff} - \tilde{f}_{eff} - f_{ext}, \\
ikM_{eff} &= \omega^2 \bar{I}_0 \psi_{eff} + F_{eff} + \tilde{q}_{eff} + q_{ext},
\end{aligned} \tag{R10}$$

where \tilde{p}_{eff} , \tilde{q}_{eff} , \tilde{s}_{eff} and \tilde{f}_{eff} are effective sources of the bending curvature, body torque, shear strain and transverse body force caused by the scatterers, respectively. For the effective medium (Fig. R3c), conservations of translational and rotational momentums and kinematic equations in the spectrum domain are

$$\begin{aligned}
ik\psi_{eff} &= \kappa_{eff} + p_{ext}, \\
ikW_{eff} &= \gamma_{eff} + \psi_{eff} + s_{ext}, \\
ikF_{ext} &= \dot{\mu}_{eff} - f_{ext}, \\
ikM_{eff} &= -\dot{J}_{eff} + F_{eff} + q_{ext}.
\end{aligned} \tag{R11}$$

Comparing Eqs. (R10) and (R11), effective translational and rotational momentums, effective curvature and effective shear strain can be written

$$\begin{aligned}
\kappa_{eff} &= -\bar{D}_0 M_{eff} + \tilde{p}_{eff}, \\
J_{eff} &= i\omega \bar{I}_0 \psi_{eff} - \frac{\tilde{q}_{eff}}{i\omega}, \\
\gamma_{eff} &= \bar{g}_0 F_{eff} + \tilde{s}_{eff}, \\
\mu_{eff} &= i\omega \bar{\rho}_0 W_{eff} - \tilde{f}_{eff}.
\end{aligned} \tag{R12}$$

The purpose next is to find analytical formulas of \tilde{p}_{eff} , \tilde{q}_{eff} , \tilde{s}_{eff} and \tilde{f}_{eff} in terms of M_{eff} , ψ_{eff} , F_{eff} and W_{eff} . Multiple scattering effects are considered for this purpose.

Based on the retrieved polarizability tensor, sources of the bending curvature and shear strain in the multipole vector of each of the scatterers are

$$\begin{aligned}
\tilde{p}_0 &= \beta_{44} M_{loc}, \\
\tilde{s}_0 &= \beta_{34} M_{loc}.
\end{aligned} \tag{R13}$$

The scattered wave fields at $x = x_m$ due to the n -th scatterer ($x = x_n$) can be written

$$M_s(x_m) = G_{pM}^{mn} \tilde{p}_n + G_{sM}^{mn} \tilde{s}_n, \tag{R14}$$

where $G_{pM}^{mn} = R_{Mw} G_p(x_m, x_n)$ and $G_{sM}^{mn} = R_{Mw} G_s(x_m, x_n)$ with R_{Mw} being the ratio of the bending moment and transverse displacement. Green's functions $G_p(x_m, x_n)$ and $G_s(x_m, x_n)$ are defined in the Supplementary Information.

As a result, the local wave fields at the zeroth scatterer can be superposed as

$$M_{loc} = M_{ext} + \sum_{n \neq 0} (G_{pM}^{0n} \tilde{p}_n + G_{sM}^{0n} \tilde{s}_n). \tag{R15}$$

We rewrite Eq. (R15) into the following compact form

$$M_{loc} = M_{ext} + C_{pM} \tilde{p}_0 + C_{sM} \tilde{s}_0, \tag{R16}$$

where

$$C_{pM} = \sum_{n \neq 0} e^{ikx_n} G_{pM}^{0n},$$

$$C_{sM} = \sum_{n \neq 0} e^{ikx_n} G_{sM}^{0n},$$

where C_{pM} and C_{sM} are known as symmetric and antisymmetric lattice sums, respectively. Sources of the effective bending curvature and shear strain can be related to microscopic responses using spatial averages: $\tilde{p}_{eff} = \frac{\tilde{p}_0}{L}$ and $\tilde{s}_{eff} = \frac{\tilde{s}_0}{L}$ with L being the lattice constant of the inhomogeneous beam. Combining Eqs. (R8), (R10), (R12), (R13) and (R16) gives the effective constitutive relations of the inhomogeneous beam

$$\begin{aligned} \kappa_{eff} &= -\left(\bar{D}_0 - \frac{\beta_{44}\Gamma}{L\Pi}\right)M_{eff}, \\ J_{eff} &= \bar{I}_0\dot{\psi}_{eff}, \\ \gamma_{eff} &= \bar{g}_0F_{eff} - \frac{\beta_{34}\Gamma\nu}{i\omega L\Pi}V_{eff} \\ \mu_{eff} &= \bar{\rho}_0V_{eff}, \end{aligned} \tag{R17}$$

where $\Gamma = \frac{-k^4 + \bar{D}_0\omega^2\bar{\rho}_0 + \bar{D}_0\bar{I}_0k^2\omega^2 + \bar{g}_0k^2\omega^2\bar{\rho}_0 - \bar{D}_0\bar{I}_0\bar{g}_0\omega^4\bar{\rho}_0}{ik\beta_{dM}\omega^2\bar{\rho}_0 + \frac{\beta_{mM}\omega^2\bar{\rho}_0}{L\Pi} + \bar{D}_0\omega^2\bar{\rho}_0 - k^4 + \bar{D}_0\bar{I}_0k^2\omega^2} + \frac{\bar{I}_0\beta_{mM}k^2\omega^2}{L\Pi} + \bar{g}_0k^2\omega^2\bar{\rho}_0 + \bar{D}_0\bar{I}_0\bar{g}_0\omega^4\bar{\rho}_0 + \frac{\bar{I}_0\beta_{mM}\bar{g}_0\omega^4\bar{\rho}_0}{L\Pi}$, $\Pi = 1 - C_{Md}\beta_{dM} - C_{Mm}\beta_{mM}$

and $\nu = \frac{W_{eff}}{M_{eff}}$ being the ratio of the effective displacement and bending moment of the corresponding mode. Equation (R17) displays the Willis coupling of the effective beam between the transverse velocity and shear force. Comparing with Eq. (R6), the effective material parameters of the Willis beam can be obtained as

$$\begin{aligned}
\bar{\rho} &= \bar{\rho}_0, \\
\bar{I} &= \bar{I}_0, \\
\bar{D} &= \bar{D}_0 - \frac{\beta_{44}\Gamma}{L\Pi}, \\
\bar{g} &= \bar{g}_0, \\
\bar{S}_{11}^{(b)} &= 0, \\
\bar{S}_{11}^{(h)} &= 0, \\
\bar{S}_{53}^{(b)} &= \frac{\beta_{34}\Gamma\nu}{i\omega L\Pi}, \\
\bar{S}_{35}^{(h)} &= 0.
\end{aligned} \tag{R18}$$

Equation (R18) represents the analytical connection between the polarizability tensor and the constitutive properties of Willis beams.

As stated before, in the revised version, we mention its importance and conclusion of the source-driven homogenization model in the last section, and move all the details into Supplementary Information. We hope the reviewer can satisfy these changes.

(C) The authors refer to the symmetry property of the Willis coupling in the abstract and the body of the text. I find this statement as ambiguous: initially I understood it as a symmetry with respect to the components of the Willis coupling, but in the text it turns out it refers to the loss of reciprocity [3], or, equivalently, the loss of self-adjointness of effective constitutive operator [4]. This should be clarified.

Response: In 3D elastodynamics, reciprocity places a restriction on Willis coupling coefficients as [Ref. 3, Eq. (3.10c)]

$$\tilde{S}_{ijk} = S_{jki}. \tag{R19}$$

When reducing Eq. (R19) to a 1D Timoshenko beam, this restriction becomes

$$\bar{S}_{11}^{(b)} = \bar{S}_{11}^{(h)}, \quad \bar{S}_{53}^{(b)} = \bar{S}_{35}^{(h)}. \tag{R20}$$

If Willis coupling coefficients of the beam satisfy Eq. (R20), we call them symmetric. Therefore, for the Timoshenko beam, loss of reciprocity is related to the symmetry property of the Willis coupling coefficients.

For this study, we consider flexural wave propagations along a beam that are embedded with a periodic arrangement of the proposed mechanical Willis scatterers having reciprocity breaking functions. Applying the source-driven homogenization model, this arrangement will lead to a beam whose constitutive relations are of the Willis form (R18). It can be clearly seen that the effective mass density, rotational inertia and shear compliance are left unchanged, and three of the Willis coupling coefficients $\bar{S}_{11}^{(b)}$, $\bar{S}_{11}^{(h)}$ and $\bar{S}_{35}^{(h)}$ are zero. Therefore, the nonzero Willis coupling coefficient, $\bar{S}_{53}^{(b)}$ ($\bar{S}_{53}^{(b)} \neq \bar{S}_{35}^{(h)}$), makes the Willis coupling asymmetry and the Willis beam lose reciprocity.

To eliminate this ambiguous, in the revised manuscript, we remove some statements on the symmetry of Willis coupling coefficients before this context appears.

(D) The authors claim that “It is also important to mention that the scattering properties of a Willis scatterer cannot be properly captured using simple the Euler beam theory because it lacks the degree of freedom associated shearing motion within the beam that is required to capture monopole-dipole coupling that is the finger-print of the Willis material response”. Why an Euler beam whose bending moment depends (not only on the curvature but also) on the transverse velocity, and whose linear momentum depends (not only on the transverse velocity but also) on the curvature [5] is not considered a beam with a Willis response?

Response: Firstly, as shown in Eq. (R6), Willis couplings in a beam can only exist between bending moment and rotational velocity and between shear force and transverse velocity. All these Willis couplings display symmetric and antisymmetric couplings.

Secondly, in fact, the Euler beam is a special case in what we discussed in Eqs. (R2) – (R6), by assuming the shear modulus to infinity. In this way, we will lose the independent rotational angle and shear force. The transverse velocity and bending moment of an Euler beam left are monopole and multipole quantities (see the response in (B)), which can only radiate waves in-phase (symmetric) to the two directions. Therefore, the couplings between them cannot describe symmetric-antisymmetric couplings or monopole-dipole couplings, which, nevertheless, are essential for Willis couplings. For this reason, Timoshenko beam theory is applied to study Willis coupling in beams.

Brief discussion has been added in Page 10 of the revised manuscript.

(E) Some of the statements made in the text should be toned down. For example, in the abstract the authors claim that “Willis coupling coefficients in Willis solids designed so far are symmetric”. However, there are several works who already broke reciprocity in Willis materials theoretically [6] and experimentally [7] (I think that the last author also has a relevant theoretical work on the topic [8]).

Another statement that I find inexact is that “In summary, we have provided a theoretic and experimental study on an active Willis meta-layer in elastic plates...”. The theoretical analysis is completely a one-dimensional analysis, and therefore is not applicable to plates, which are governed by equations of two spatial coordinates. In the SI, the authors themselves say that “Note that the approach demonstrated below can be easily extended to a Willis plate through the

use of Mindlin–Reissner plate theory”, i.e., they did not execute this extension. I suggest that the authors would either extend their approach as they say, or rephrase their statements.

Response: According to the reviewer’s suggestion, we have toned down some statements made in the text and modified the sentence in Abstract and Conclusions as

“However, Willis coupling coefficients designed so far are intrinsically coupled and strongly frequency-dependent, which inhibits their full implementation in structural dynamic applications.”

and

“In summary, we have provided a theoretical and experimental study of an active Willis meta-layer using a sensor-actuator control loop to realize independently controlled asymmetric polarizabilities.”

(F) Finally, the manuscript should undergo a thorough proofreading, as it contains several typos (e.g., line 68 is missing the word “to” after “ability”, in line 181 the words “is” and “leads” should be “are” and “lead” since they refer to the word “functions”).

Response: The manuscript has gone through proofreading by all authors. We have changed those typos pointed by the reviewer.

References

[1] C. F. Sieck, A. Alù, and M. R. Haberman, *Phys. Rev. B* 96, 104303 (2017). □

[2] X. Su and A. N. Norris, *Phys. Rev. B* 98, 174305 (2018)

[3] M. B. Muhlestein, C. F. Sieck, A. Alù, and M. R. Haberman, *Proc. R. Soc. A* 472, 20160604 (2016)

[4] J. R. Willis, *Proc. R. Soc. A* 467, 1865 (2011)

- [5] R. P. Salomon, G. Shmuel, *J. Mech. Phys. Solids* 119 (2018)
- [6] L. Quan, D. L. Sounas, and A. Alu, *Phys. Rev. Lett.*, 123, 064301(2019)
- [7] Y. Zhai, H.-S. Kwon, and B.-I. Popa, *Phys. Rev. B* 99, 220301(R) (2019)
- [8] H. Nassar, X. Xu, A. Norris, and G. Huang, *J. Mech. Phys. Solids* 101, 10 (2017)

REVIEWER COMMENTS

Reviewer #1 (Remarks to the Author):

The revised manuscript is greatly improved, and gives more appropriate emphasis on the important experimental aspects. Also, the multipole model is now much better justified. The manuscript should be suitable for publication, once the following issues are clarified:

On line 237, the reader is referred to the Supplementary Information for interpretation of β_{34} and β_{44} . While it makes sense that the full definition should be kept in the Supplementary Information, these coefficients should be given some name and interpretation to help the reader understand their significance. To my understanding these represent "bending to shear" and "bending to bending" terms respectively.

Now that it has been clarified that p_0 and M_{loc} are quadrupole terms, and that only bending and shear terms are significant in the structure, it seems that the Willis coupling is in the quadrupole-dipole terms. As far as I know, all Willis coupling reported to date in acoustics is for monopole-dipole coupling, so this difference should be mentioned in the manuscript.

It is not clear that the claims (including in the abstract) about being broadband and overcoming strong frequency-dependence are fully justified. My understanding of Figure 3 (e) is that results for each frequency were obtained by reconfiguring the transfer function each time. If this understanding is correct, then the structure is not truly broadband, in the sense of being able to fully manipulate a short pulse, so claims of being broadband should be toned down (e.g. state only that the structure is reconfigurable over a broad bandwidth). The key theoretical point, which the authors may wish to discuss, is whether the functions H_1 and H_2 given in Eq (1) correspond to causal functions in the time-domain.

I am still not convinced that the section on Homogenization really adds any value to the paper. On the other hand, it takes up minimal space in the revised manuscript.

Minor points:

- The order of supplementary Figure S5 (c) and (d) should be swapped, so that it corresponds to the order these terms appear in Q

Reviewer #2 (Remarks to the Author):

The revision has significantly improved the manuscript, and most of the points made were addressed. I disagree, however, with the reply to point (D). In the first part of the reply, the authors claim that Eq. (R6) implies that Willis couplings cannot exist between the bending moment and the velocity, nor between the linear momentum and the curvature; this claim is inconsistent with Ref. [5], which shows the converse. The issue with Eq. (R6) is that it is obtained by assuming a single constituent whose constitutive response is described by a Willis material, and then applying the standard procedure from beam theory to obtain equations in terms of forces, bending moments etc. In a sense, it carries out an homogenization from the equations of (Willis) elastodynamics to the equations of the mechanics of structures. However, Eq. (R6) does not model the homogenized constitutive relation of a heterogeneous beam whose structural properties vary in space, and the arguments that are based on it in the second part of the reply do not apply in this case. In accordance, Eq. (R10) is an assumption, and not the end of a homogenization process at the structural level. The inconsistency with the analysis based on the source-driven homogenization used by the authors may be caused by the fact that the body force density is missing; the transverse body force they consider (f in the notation of the manuscript) is a higher order moment of it. It is this density that appears in the beam governing equation. Its

analogue in elastodynamics is the one that is prescribed as the source in the literature, see, e.g., [4,8], in the same way that the analogue of the bending curvature source appears in [4,8], referred to as an "eigen-strain". In summary, the authors should either disprove the arguments above, or remove the statement that an Euler beam cannot exhibit Willis coupling.

Response to reviewer's comments (#1)

The authors would like to take this opportunity to thank the reviewer for his/her insights and suggestions, and have responded to the comments below. The paper has been modified accordingly.

The revised manuscript is greatly improved, and gives more appropriate emphasis on the important experimental aspects. Also, the multipole model is now much better justified. The manuscript should be suitable for publication, once the following issues are clarified:

Response: We thank the reviewer for the positive comments on the first revision. We have clarified the following issues point-by-point below.

(A) On line 237, the reader is referred to the Supplementary Information for interpretation of β'_{34} and β'_{44} . While it makes sense that the full definition should be kept in the Supplementary Information, these coefficients should be given some name and interpretation to help the reader understand their significance. To my understanding these represent "bending to shear" and "bending to bending" terms respectively.

Response: We appreciate the reviewer's suggestion to highlight the importance and meaning of terms β'_{34} and β'_{44} in the polarizability tensor, which the reviewer correctly interpreted. We thus added the following clarifying sentence to the revised manuscript:

"In particular, β'_{34} describes the coupling between the quadrupole bending moment and the dipole shear strain, and β'_{44} represents the coupling between the quadrupole bending moment and the quadrupole bending curvature. Thus, β'_{34} and β'_{44} can be termed as quadrupole-to-dipole and quadrupole-to-quadrupole polarizabilities, respectively."

(B) Now that it has been clarified that p_0 and M_{loc} are quadrupole terms, and that only bending and shear terms are significant in the structure, it seems that the Willis coupling is in the

quadrupole-dipole terms. As far as I know, all Willis coupling reported to date in acoustics is for monopole-dipole coupling, so this difference should be mentioned in the manuscript.

Response: We took this excellent suggestion too in the revised manuscript, which is shown below:

“It should be noted that Willis coupling reported in acoustics and elastodynamics generally only considers monopole-dipole coupling. However, the Willis coupling in the meta-layer presented in this work exploits coupling between higher order terms, specifically quadrupole-to-dipole and quadrupole-to-quadrupole couplings.”

To precisely reflect this contribution on Willis coupling, the phrase “higher-order” has been also added into the abstract.

(C) It is not clear that the claims (including in the abstract) about being broadband and overcoming strong frequency-dependence are fully justified. My understanding of Figure 3 (e) is that results for each frequency were obtained by reconfiguring the transfer function each time. If this understanding is correct, then the structure is not truly broadband, in the sense of being able to fully manipulate a short pulse, so claims of being broadband should be toned down (e.g. state only that the structure is reconfigurable over a broad bandwidth). The key theoretical point, which the authors may wish to discuss, is whether the functions H_1 and H_2 given in Eq (1) correspond to causal functions in the time-domain.

Response: The understanding of the reviewer is correct. To obtain the broadband results in Fig. 3e, the transfer function must be reconfigured. According to the reviewer’s suggestion, we changed the statement on “broadband operability” to “reconfigurable over a broad bandwidth” in the revised manuscript to more precisely describe the operation of the meta-layer.

Frequency responses of transfer functions, H_1 and H_2 , given in Eq. (1), are derived from frequency-domain analyses, such that causality may not be guaranteed in time domain. In the experiments, we made use of two sixth-order low-pass infinite impulse response (IIR) filters to construct the two transfer functions. To achieve the frequency responses described in Eq. (1) at each of the single frequencies, we finely tuned the amplification ratio of the filters and added

proper time delay on the output signals during each of the reconfiguration steps. It can be clearly seen from experiments that the linear signal processing filters in the time domain are inevitably causal. We added the discussion of this point in the revision of both the main text and Supplementary Information (see Source Data for the coefficients of the filter).

(D) I am still not convinced that the section on Homogenization really adds any value to the paper. On the other hand, it takes up minimal space in the revised manuscript.

Response: We agree that the section on homogenization is not the primary contribution of the paper and we have therefore minimized the space dedicated to this section. However, we feel that this portion of the paper provides a succinct discussion regarding the extension of the observed behavior of the active scatterer meta-layer to general cases, specifically nonreciprocal 1D and 2D Willis media in structural elements like beams and plates. The homogenization method we derived in the Supplementary Information provides interested readers with the formal analytical procedure following a source driven homogenization formalism to determine the effective parameters based on the polarizabilities of the active scatterers up to quadrupolar order. We also note that the changes made for this revision address the insightful comments raised by Reviewer #2.

(E) Minor points:

- The order of supplementary Figure S5 (c) and (d) should be swapped, so that it corresponds to the order these terms appear in Q.

Response: Supplementary Figures S5 (c) and (d) have been swapped in the Supplementary Information.

Response to reviewer's comments (#2)

The authors would like to take this opportunity to thank the reviewer for his/her insights and suggestions, and have responded to the comments below. The paper has been modified accordingly.

(A) The revision has significantly improved the manuscript, and most of the points made were addressed. I disagree, however, with the reply to point (D). In the first part of the reply, the authors claim that Eq. (R6) implies that Willis couplings cannot exist between the bending moment and the velocity, nor between the linear momentum and the curvature; this claim is inconsistent with Ref. [5], which shows the converse. The issue with Eq. (R6) is that it is obtained by assuming a single constituent whose constitutive response is described by a Willis material, and then applying the standard procedure from beam theory to obtain equations in terms of forces, bending moments etc. In a sense, it carries out an homogenization from the equations of (Willis) elastodynamics to the equations of the mechanics of structures. However, Eq. (R6) does not model the homogenized constitutive relation of a heterogeneous beam whose structural properties vary in space, and the arguments that are based on it in the second part of the reply do not apply in this case. In accordance, Eq. (R10) is an assumption, and not the end of a homogenization process at the structural level. The inconsistency with the analysis based on the source-driven homogenization used by the authors may be caused by the fact that the body force density is missing; the transverse body force they consider (f in the notation of the manuscript) is a higher order moment of it. It is this density that appears in the beam governing equation. Its analogue in elastodynamics is the one that is prescribed as the source in the literature, see, e.g., [4,8], in the same way that the analogue of the bending curvature source appears in [4,8], referred to as an “eigen-strain”. In summary, the authors should either disprove the arguments above, or remove the statement that an Euler beam cannot exhibit Willis coupling.

Response: We thank the reviewer for their insightful comments on point (D) in the first revision. We agree with the reviewer on the discussion about homogenization approaches for beams. Our approach in Eqs. (R2) – (R6) does indeed presuppose the equations of (Willis) elastodynamics and specialize them to the equations of motion for the mechanics of structures. Equation (R6) can therefore only capture the Willis coupling from the homogenized medium when the stress,

strain, particle velocity and linear momentum fields are already averaged. As a result, in our initial submission and in the first revision, we found that eight terms in the constitutive matrix of Eq. (R6) are equal to zero, and thus that the coupling between the bending moment and the velocity, and between the linear momentum and the curvature were missing. Thanks to reviewer's insight and suggestions, the revised manuscript now starts with a general homogenization based on heterogeneous Timoshenko beam equations (following the approach of Ref. [27] in the revised main article file). As a result, coupling between bending moment, shear stress, transverse velocity, rotational angle, bending curvature, shear strain, as well as translational and rotational momentums is included in the general constitutive relations of Willis beams.

The insight from the second reviewer has therefore prompted us to revisit the homogenization section in the main text and the Supplementary Information (SI). At a top-level, we have made following changes: (1) We have removed the statement that an Euler beam cannot exhibit Willis coupling; (2) We have rewritten the section on "Material parameters of Willis beams" by deleting Eqs. (22) – (26) in Supplementary Information, and adding discussions on the general constitutive relations of Willis beams; and (3) We have modified the source-driven homogenization approach based on the new general form.

In particular,

(1) The original statement on Page 10 in the main text "It is also important to mention that the scattering properties of a Willis scatterer cannot be properly captured using the simple Euler beam theory because it lacks the degrees of freedom associated shearing motion within the beam and thus can only capture symmetric-to-symmetric coupling. Timoshenko beam theory is therefore employed in order to capture symmetric-to-antisymmetric (monopole-dipole or multipole-dipole) coupling that is the finger-print of the Willis material response." has been changed as

"It is also important to mention that the scattering properties of the Willis scatterer presented here cannot be properly captured using the simple Euler beam theory because it lacks the degrees of freedom associated shearing motion within the beam. We therefore employ Timoshenko beam theory."

(2) The Section “Material parameters of Willis beams” in the Supplementary Information has been modified as

“Following homogenization approach in Ref [27], we suggest a general homogenization approach within the framework of Timoshenko beam theory by defining the general constitutive relation of Willis beams in a matrix form as

$$\begin{bmatrix} \kappa \\ J \\ \gamma \\ \mu \end{bmatrix} = \begin{bmatrix} -\bar{D} & S_{12} & S_{13} & S_{14} \\ S_{21} & \bar{I} & S_{23} & S_{24} \\ S_{31} & S_{32} & \bar{g} & S_{34} \\ S_{41} & S_{42} & S_{43} & \bar{\rho} \end{bmatrix} \begin{bmatrix} M \\ \psi \\ F \\ V \end{bmatrix}, \quad (\text{R1})$$

where M , ψ , F and V denote the bending moment, rotational angle of the beam section, shear force, and transverse velocity, respectively, and κ , J , γ and μ represent the bending curvature, angular momentum, shear strain and linear momentum, respectively. In Eq. (R1), \bar{D} , \bar{I} , \bar{g} and $\bar{\rho}$ are the effective bending compliance, mass moment of inertia, inverse of the shear modulus, and mass density of the heterogeneous beam, respectively. The off-diagonal terms, S_{ij} ($i \neq j$), in the matrix denote the coupling coefficients of a Willis beam using Timoshenko theory. The Willis coupling in beams differs from that of 3D linear elastodynamic Willis constitutive form in that it exhibits coupling between the higher order stress, bending curvature, local rotational angle and angular momentum.”

(3) The Section “Source-driven homogenization theory” in the Supplementary Information has also been modified accordingly as

“Upon application of the source-driven homogenization procedure, the effective constitutive relations for a microscopically heterogeneous beam consisting of a background elastic beam and an array of periodic scatters that couple symmetric and antisymmetric multipoles as proposed in the main text can be written as

$$\begin{aligned}
\kappa_{eff} &= -\left(\bar{D}_0 - \frac{\beta_{44}\Gamma}{L\Pi}\right)M_{eff}, \\
J_{eff} &= \bar{I}_0\dot{\psi}_{eff}, \\
\gamma_{eff} &= \bar{g}_0F_{eff} - \frac{\beta_{34}\Gamma}{L\Pi}M_{eff}, \\
\mu_{eff} &= \bar{\rho}_0V_{eff},
\end{aligned} \tag{R2}$$

where

$$\begin{aligned}
\Gamma &= \frac{-k^4 + \bar{D}_0\omega^2\bar{\rho}_0 + \bar{D}_0\bar{I}_0k^2\omega^2 + \bar{g}_0k^2\omega^2\bar{\rho}_0 - \bar{D}_0\bar{I}_0\bar{g}_0\omega^4\bar{\rho}_0}{\frac{ik\beta_{dM}\omega^2\bar{\rho}_0}{L\Pi} + \frac{\beta_{mM}\omega^2\bar{\rho}_0}{L\Pi} + \bar{D}_0\omega^2\bar{\rho}_0 - k^4 + \bar{D}_0\bar{I}_0k^2\omega^2} \\
&\quad + \frac{\bar{I}_0\beta_{mM}k^2\omega^2}{L\Pi} + \bar{g}_0k^2\omega^2\bar{\rho}_0 + \bar{D}_0\bar{I}_0\bar{g}_0\omega^4\bar{\rho}_0 + \frac{\bar{I}_0\beta_{mM}\bar{g}_0\omega^4\bar{\rho}_0}{L\Pi}, \\
\Pi &= 1 - C_{Md}\beta_{dM} - C_{Mm}\beta_{mM}.
\end{aligned}$$

The third equation in Eq. (R2) displays the Willis coupling of the homogenized Willis beam, where the bending moment (symmetric) and shear strain (antisymmetric) are coupled. Comparing with Eq. (R1), the effective material parameters of the Willis beam can be identified by matching terms:

$$\begin{aligned}
\bar{D} &= \bar{D}_0 - \frac{\beta_{44}\Gamma}{L\Pi}, \\
\bar{I} &= \bar{I}_0, \\
\bar{g} &= \bar{g}_0, \\
\bar{\rho} &= \bar{\rho}_0, \\
S_{31} &= -\frac{\beta_{34}\Gamma}{L\Pi}, \\
S_{ij} &= 0, \text{ for others.}
\end{aligned} \tag{R3}$$

Equation (R3) shows the analytical form of the connection between the polarizability tensor of the active scatterers considered in this work and the resulting effective constitutive properties of Willis beams using Timoshenko theory. From Eq. (R3), it can be clearly seen that the effective mass density, rotational inertia and shear compliance are left unchanged, and only one Willis coupling coefficient is nonzero. Applying conservation of translational and rotational momentum

($\dot{\mu} = \frac{\partial F}{\partial x} + f$, $\dot{J} = \frac{\partial M}{\partial x} - F + m$), we find that the coupling coefficient, S_{31} , will induce nonreciprocal wave propagation in the periodic Willis beam considered in this work. On the other hand, the effective bending stiffness of the Willis beam is modified by the polarizability, β_{44} , which is reciprocal.”

In summary, we believe the converse is mainly due to the fact that the constitutive forms from different homogenization approaches (within Euler or Timoshenko beam theory) are usually non-unique. Following the reviewer’s suggestion to resolve the inconsistency in our proposed method with that of Ref. [27], we begin with the general constitutive form of Willis beam that assumes coupling between the first four multipoles of the scatterer polarizability and use a source-driven method to retrieve the effective properties of Willis beam. Furthermore, we note that Eq. (R1) is now consistent with Ref. [27]. We also check the formulation of the source-driven method and believe that the body force density term \tilde{f}_{eff} in Eq. (R10) is separated from $\tilde{f}(x) = [\omega^2 \bar{\rho}(x) - \omega^2 \bar{\rho}_0] W(x)$ and indeed the effective body force density.

REVIEWERS' COMMENTS:

Reviewer #1 (Remarks to the Author):

I am satisfied that all necessary changes have been made to this manuscript, and it is now suitable for publication.

Reviewer #2 (Remarks to the Author):

The authors have addressed the remaining point in the revised version. I suggest to accepted the manuscript for publication.

Reviewer #1 (Remarks to the Author):

I am satisfied that all necessary changes have been made to this manuscript, and it is now suitable for publication.

Reviewer #2 (Remarks to the Author):

The authors have addressed the remaining point in the revised version. I suggest to accepted the manuscript for publication.

Response: The authors would like to thank the reviewers for his/her insights and suggestions for improving the manuscript.